# An alpha-helical lid guides the target DNA toward catalysis in CRISPR-Cas12a

Aakash Saha [1], Mohd Ahsan [1,4], Pablo R. Arantes[1,4], Michael Schmitz [2], Christelle Chanez [2], Martin Jinek [2] & Giulia Palermo [1,3] ✉

CRISPR-Cas12a is a powerful RNA-guided genome-editing system that generates double-strand DNA breaks using its single RuvC nuclease domain by a sequential mechanism in which initial cleavage of the non-target strand is followed by target strand cleavage. How the spatially distant DNA target strand traverses toward the RuvC catalytic core is presently not understood. Here, continuous tens of microsecond-long molecular dynamics and free-energy simulations reveal that an α-helical lid, located within the RuvC domain, plays a pivotal role in the traversal of the DNA target strand by anchoring the crRNA:target strand duplex and guiding the target strand toward the RuvC core, as also corroborated by DNA cleavage experiments. In this mechanism, the REC2 domain pushes the crRNA:target strand duplex toward the core of the enzyme, while the Nuc domain aids the bending and accommodation of the target strand within the RuvC core by bending inward. Understanding of this critical process underlying Cas12a activity will enrich fundamental knowledge and facilitate further engineering strategies for genome editing.

Bacteria and other prokaryotes deploy an arsenal of CRISPR-Cas systems (clustered regularly interspaced short palindromic repeats, CRISPR-associated proteins) as a part of their adaptive immune response against foreign nucleic acid invasion[1]. The CRISPR-associated proteins Cas9[2] and Cas12a[3], being RNA-programmable endonucleases, launched unparalleled genome-editing applications across different fields of biotechnology, agriculture, and medicine[4,5]. The Cas12a nuclease came to the forefront with innovative applications such as extraordinarily rapid nucleic acid detection[6], also harnessed for the detection of the SARS-CoV-2[7,8]. At the molecular level, Cas12a cleaves double-stranded target DNA using a single catalytic domain[9–14]. Hence, conformational changes of the protein are critical for the accommodation and cleavage of DNA, but the mechanism of such a conformational transition is highly elusive.

Structures of Cas12a from *Francisella novicida* (FnCas12a) revealed a bilobed architecture, comprising of a recognition (REC) and a nuclease (NUC) lobe, flanked by the wedge (WED) domain (Fig. 1)[9–12]. Two alpha-helical domains (REC1 and REC2) constitute the REC lobe and mediate nucleic acid binding, while the NUC lobe comprises the RuvC catalytic domain and an adjacent Nuc domain. During its biophysical function, Cas12a uses CRISPR RNA (crRNA) for highly specific molecular recognition[15–17] and cleavage of complementary DNA sequences. Upon recognition and binding of a short protospacer-adjacent motif (PAM) in the target DNA by the PAM-interacting (PI) domain of Cas12a, the crRNA binds one DNA strand (i.e., the target strand, TS) and forms a crRNA:TS duplex, while the other non-target strand (NTS) gets displaced. Cas12a then confers double-stranded DNA breaks by using only the RuvC domain as its molecular scissor. This is strikingly different from Cas9, where two catalytic domains – HNH and RuvC – perform cleavage of the TS and NTS, respectively[18]. This difference alludes to a conundrum of how a single nuclease domain could enable cleavage of both DNA strands in Cas12a. Indeed, while the NTS accommodates within the RuvC catalytic cleft, the TS stays spatially distant from the catalytic pocket, close to the REC lobe (Fig. 1). Experimental observations have inferred that the TS gets cleaved post the cleavage of the NTS[9,12,19]. Together, this indicates a conformational

[1]Department of Bioengineering, University of California Riverside, 900 University Avenue, Riverside, CA 52512, USA. [2]Department of Biochemistry, University of Zürich, Winterthurerstrasse 190, CH-8057 Zürich, Switzerland. [3]Department of Chemistry, University of California Riverside, 900 University Avenue, Riverside, CA 52512, USA. [4]These authors contributed equally: Mohd Ahsan, Pablo R. Arantes. ✉e-mail: giulia.palermo@ucr.edu

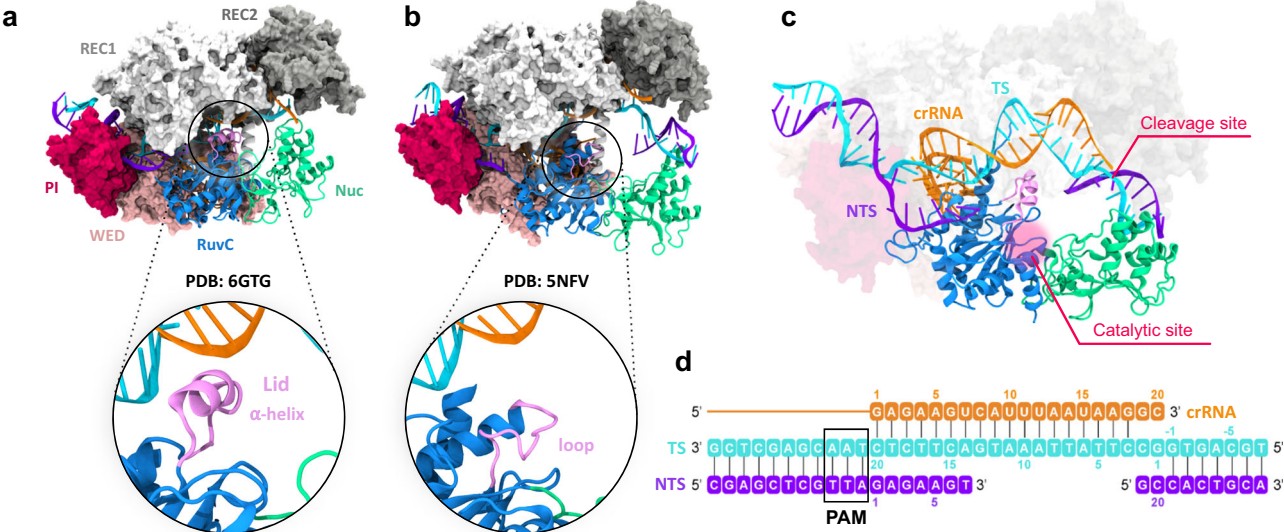

**Fig. 1 | Structures of CRISPR-Cas12a after DNA non-target strand cleavage.** The Cas12a protein is shown in molecular surface, highlighting the RuvC (blue) and Nuc (green) domains as ribbons. The crRNA (orange), the DNA target strand (TS, cyan) and the non-target strand (NTS, violet) are also shown as ribbons. **a** The cryo-EM structure (PDB: 6GTG (Cas12a – I4 Conformation))[12] contains a short TS without the scissile phosphate. This structure displays an α-helical "lid", opening the RuvC domain (close-up view). **b** The X-ray structure (PDB: 5NFV (FnCas12a bound to R-loop))[9] reports a longer TS, including the scissile phosphate and reconciling with the NTS. Here, the lid is unstructured (close-up view) and reconstructed through homology modeling. **c** A more complete CRISPR-Cas12a modeled from the cryo-EM (**a**) and X-ray (**b**) structures, containing the α-helical lid and a longer DNA TS including the cleavage site. **d** Schematic representation of the nucleic acids.

change after the NTS cleavage, allowing the accommodation of the TS within the RuvC catalytic pocket. Nevertheless, the molecular details of these intriguing biophysical processes have remained incompletely understood.

Alongside biochemical and structural studies[9,12,20], single-molecule experiments[12,16,19,21–24] have substantially contributed toward delineating the biophysical functioning of the Cas12a enzyme. The RuvC domain is structurally stable in Cas12a, while the REC2 and Nuc domains display increased flexibility and concerted motions upon DNA binding, which could favor the conformational transition of the TS toward the catalytic site. Such cooperative movements of REC2 and Nuc have been demonstrated by cryo-EM and single-molecule FRET experiments[12], and shown through molecular simulations[25]. However, the protein dynamics and the conformational transitions orchestrating the traversal of the TS from the REC lobe to the RuvC catalytic site are yet to be identified. This is a critical question, understanding of which can lay the ground for further engineering toward improved genome editing.

Here, extensive Molecular Dynamics (MD) and free-energy simulations of Cas12a are used to unravel the process of DNA TS traversal and its repositioning within the RuvC catalytic core from a total simulation time of ~170 μs. Using Anton2, one of the fastest super-computers available for MD simulations[26], we performed continuous tens of μs-long runs, identifying an α-helical "lid" as a key structural element, and the REC2 and Nuc domains as the major players orchestrating the conformational transition. Next, free-energy methods captured the elaborate mechanism by which the α-helical lid guides the DNA TS toward the catalytic core, as also supported by DNA cleavage experiments. Taken together, our findings shed light on a cardinal step in the functioning of Cas12a, which has previously remained highly elusive.

## Results
### An α-helical lid anchors the crRNA:TS
Our investigations considered two structures obtained through cryo-EM and crystallographic techniques upon NTS cleavage with different lengths of DNA TS (Supplementary Table 1). The cryo-EM structure (PDB: 6GTG (Cas12a – I4 Conformation); EMD-0065[12], Fig. 1a) contains

a short TS without the scissile phosphate. This structure has an open RuvC domain, displaying an α-helical "lid" (residues L1008–K1021) that was held accountable for opening the RuvC domain for DNA accommodation[12]. The X-ray structure (PDB: 5NFV (FnCas12a bound to R-loop)[9], Fig. 1b) displays a full-length TS with the scissile phosphate, while the lid is structurally disordered.

Our ~10 μs long simulations of the 6GTG (Cas12a – I4 Conformation) cryo-EM structure suggest that the α-helical lid plays a crucial role in the major conformational changes of the system. Indeed, the α-helical lid draws the TS by establishing extensive interactions between its Lys/Arg residues and the crRNA:TS duplex backbone at PAM-distal sites (Supplementary Fig. 1). It is notable, however, that in this cryo-EM structure, the region of the TS downstream of the crRNA:TS duplex is absent and consequently lacks the scissile phosphate[12]. Hence, we modeled the downstream region of the DNA in the 6GTG (Cas12a – I4 Conformation) structure based on the 5NFV (FnCas12a bound to R-loop) X-ray structure[9], obtaining a more complete system of Cas12a after NTS cleavage (Fig. 1c; more details in the methods section). Two ~10 μs long replicates of this system consistently show that positively charged residues of the α-helical lid interact with the crRNA:TS duplex at PAM-distal sites after ~2 μs of MD (Fig. 2a, b). These include the stable interaction of K1013 with the TS at position 2, and R1016 with the crRNA at position 12. A bending of the TS at the end of the crRNA:TS duplex is also observed that becomes more prominent after ~5.2 μs in both simulation replicates, leading to an ~40° bending with respect to the initial state (Fig. 2c). This is an important observation, which is in line with the experimental evidence of a distortion of the TS downstream of the crRNA:TS duplex that was deemed necessary for it to reach the RuvC cleavage site[27]. Remarkably, after ~5.8 μs of MD, the region of the TS including the scissile phosphate also moves toward the RuvC active site in both MD runs (Fig. 2d), further supporting the experimental indications. It is notable that biochemical and single-molecule experiments have shown that the cleavage site on the TS can mainly be located between base positions −2 and −4[22,27]. Hence, we measured the distance between the scissile phosphate connecting the $T_{-2}$ and $G_{-3}$ nucleobases and the center of mass of the RuvC catalytic core (Fig. 2d, Supplementary Fig. 2d, e). The RuvC catalytic core is composed of the D917, E1006, and D1255 residues coordinating two

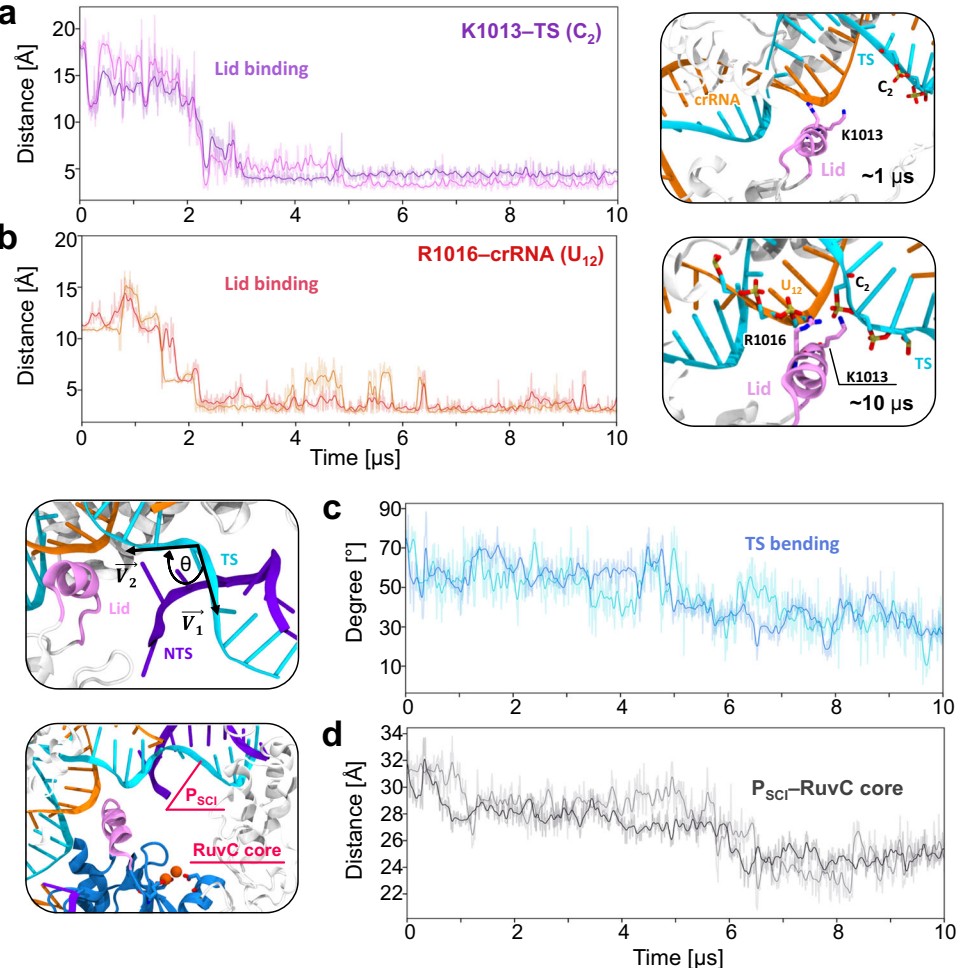

**Fig. 2 | The α-helical lid anchors the crRNA:TS duplex. a** Interactions established by the positively charged residues of the α-helical lid with the DNA target strand (K1013:NZ – $C_2$:OP1), and **b** the crRNA (R1016:NH1 – $U_{-12}$:OP1), evolving over ~10 μs of molecular dynamics simulations in two replicates. Interactions are measured as oxygen-nitrogen distance and are considered formed under 4 Å distance[66]. Representative snapshots on the right show the interactions between the α-helical lid (mauve) and the crRNA:TS duplex (crRNA: orange, TS: cyan) at different simulation times. **c** Bending of the DNA TS, computed as angle $\theta$ between the vectors $\vec{V_1}$ and $\vec{V_2}$ passing through $G_{-1}$, $G_1$, $C_2$, $C_3$ and $T_4$ (left panel, details in the

Supplementary Methods). After ~5 μs of MD, the region of the TS at the end of the duplex bends, leading to an ~40° bending with respect to the initial state. **d** Distance between the scissile phosphate ($P_{SCI}$, between the $T_{-2}$ and $G_{-3}$ nucleobases) and the center of mass of the RuvC catalytic core (left panel, composed of D917, E1006 and D1255 residues coordinating with two $Mg^{2+}$ ions). $P_{SCI}$ moves toward the RuvC core after ~5.8 μs of MD. Data are reported for two simulation replicates of ~10 μs each. The translucent lines in the plots show the actual data points, while the solid lines show the running average over 5 ns windows of MD simulation.

$Mg^{2+}$ ions, displaying a conserved two-metal ion architecture similar to Cas9[28,29]. During our ~10 μs long replicates, a substantial decrease in the computed distance indicates that the scissile phosphate approaches the RuvC core.

Taken together, the two ~10 μs long simulations of Cas12a, including a longer TS and the scissile phosphate, confirm that the α-helical lid anchors the crRNA:TS duplex at PAM-distal sites. Upon these interactions, the TS bends and the scissile phosphate moves toward the RuvC core. The observed interactions thereby suggest that the lid could have a critical role in the traversal of the DNA TS for catalysis.

The 5NFV (FnCas12a bound to R-loop) X-ray structure captured the protein complex after NTS cleavage, without a structured lid[9]. Homology modeling reconstructed this disordered lid into an unstructured loop (Fig. 1b, see the Methods section). Molecular simulations (~10 μs in two replicates) of this system show very high flexibility of the unstructured loop, especially in comparison with the α-helical lid (Supplementary Fig. 3). Interestingly, this unstructured lid forms electrostatic interactions with the crRNA but fails to establish persistent interactions with the TS (Supplementary Fig. 4a, b). These

interactions between the lid and the crRNA:TS duplex are slightly different (adjacent bases) compared to the 6GTG (Cas12a – I4 Conformation) cryo-EM structure, mostly likely owing to the differences in the lengths of the TS. Thus, though the TS comes closer to the RuvC catalytic site (Supplementary Fig. 4d), most likely owing to the inward movement of REC2 and the arching of the crRNA-TS duplex toward RuvC (Supplementary Fig. 5), yet the TS fails to bend (Supplementary Fig. 4c). Considering that TS bending was deemed crucial for its accommodation at the RuvC active site[27], an unstructured lid is thus functionally less effective for the accommodation of the TS.

## REC2 and Nuc facilitate the traversal of the TS
Our long-timescale simulations of CRISPR-Cas12a including a complete TS (Fig. 1c) also reveal noteworthy conformational changes of the protein and the crRNA:TS duplex, suggesting a cooperative dynamical mechanism fostering the traversal of the TS. To monitor these conformational changes, we analyzed the bending angles between the vector passing through the regions of interest and its perpendicular out of the plane, with respect to the first frame of reference (details in the Supplementary Methods).

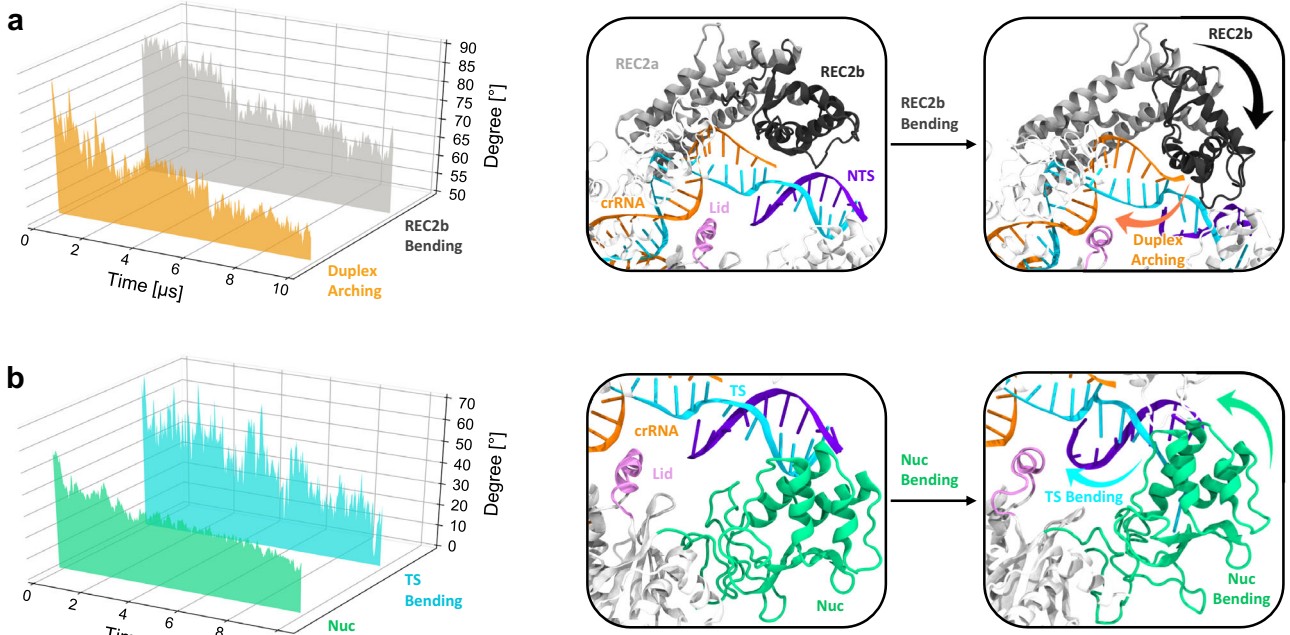

**Fig. 3 | Bending of major domains in CRISPR-Cas12a. a** Bending of the REC2b region (residues 384–473, 543–557, gray) and arching of the terminal major groove of the crRNA:TS duplex (base pairs $A_{13}$:$G_{19}$–$T_8$:$C_2$) along ~10 μs of MD simulations of CRISPR-Cas12a including a complete TS. The REC2b and crRNA:TS duplex conformational changes are shown on the right. **b** Bending of the DNA TS (cyan, Fig. 2d) in concert with the bending of the Nuc domain (green) evolving over the multi-μs simulation. The panels on the right show the conformational changes. The curved arrows indicate the conformational changes in the domains. Details are in the Supplementary Methods and Supplementary Table 2. Data are shown here for one simulation replica, and for two replicas in Supplementary Fig. 6, in which regular time-series plots are reported.

We observe that the long α-helices of REC2 binding the crRNA:TS minor groove (residues 339–383, 474–543, 557–590; hereafter referred as REC2a) exhibit restricted motions. On the other hand, the short alpha helices connected by turns (residues 384–473, 543–557; referred to as REC2b) display a marked inward movement along the two ~10 μs long simulations. This is shown by a decrement in the bending angle of REC2b (Fig. 3a, gray), which is noteworthy at ~3 μs and after ~6 μs, with REC2b becoming more bent in one simulation replica than the other after ~6 μs (as detailed in regular time-series plots in Supplementary Fig. 6a). This conformational change follows a similar trend of the arching of the crRNA:TS duplex (Fig. 3a, orange and Supplementary Fig. 6b). Indeed, the terminal major groove of the duplex, adjacent to REC2b, arches toward the RuvC domain, as shown by the reduction of its vector angle in both the simulation replicates (Supplementary Fig. 6b). The duplex arching is remarkable ~6 μs of MD onward, alongside REC2b bending. The cooperativity between the REC2b bending and hybrid arching is also shown by the positive correlation between the two variables (Supplementary Fig. 6f). Additionally, REC2b maintains extensive contacts with the crRNA:TS duplex throughout the simulations (Supplementary Fig. 7a). This indicates that the inward motion of REC2b helps the arching of the crRNA:TS duplex. Both conformational changes are observed conspicuously upon the α-helical lid binding to the crRNA:TS duplex, occurring at ~2 μs in both simulation replicates (Fig. 2). These observations suggest the cooperativity between REC2 and the α-helical lid. Indeed, the α-helical lid anchors the crRNA:TS duplex, which is followed by the bending of REC2, facilitating conformational changes of the duplex toward the RuvC catalytic pocket.

Our simulations also show that the Nuc domain bends toward the core of the enzyme, as evidenced by the decrement of the Nuc bending angle (Fig. 3b, green and Supplementary Fig. 6c reporting regular time-series plots for two simulation replicas). During this conformational transition, Nuc maintains a noteworthy number of contacts with the 5′-region of the TS (Supplementary Fig. 7b). The DNA TS at the end of the duplex also bends in both the simulation replicates (Fig. 3b, cyan and Supplementary Fig. 6d). This similar trend of the Nuc and TS bending angles, supported by a positive correlation between the two variables (Supplementary Fig. 6g), along with the persistent interactions between them suggest that the bending of Nuc aids in the bending of the DNA TS.

Cryo-EM and single-molecule experiments reported high flexibility and coupled dynamics of REC2 and Nuc, suggesting these conformational changes to be crucial for DNA cleavage[12,16,21–24]. The continuous multi-μs long simulations reported here demonstrate that REC2 and Nuc bend toward the core of the enzyme (Fig. 3 and Supplementary Fig. 6e), while establishing multiple contacts between them (Supplementary Fig. 7c). These large-scale coupled motions exhibited by the major domains, along with the lid anchoring the TS, offer a compact conformation to the whole protein, facilitating the traversal of the TS toward the catalytic pocket.

## Traversal of the DNA TS and accommodation in the RuvC catalytic pocket

The region of the TS including the scissile phosphate is significantly far from the catalytic core (~30 Å, Fig. 1), which opens an overarching question regarding its traversal and accommodation within RuvC. To investigate this phenomenon, we performed free-energy simulations, sampling the dynamics of the complex from the complete Cas12a structure (Fig. 1c) to a catalytically competent state (Supplementary Fig. 8). The latter was built based on related X-ray structures of Cas12a orthologues holding the DNA TS within the RuvC active site (see Supplementary Methods). This catalytically competent state was subjected to ~10 μs long MD simulations to ensure extensive relaxation, revealing that no further major conformational change happens in the long timescale, especially in the dynamics of REC2 and Nuc (Supplementary Fig. 9a). This is indeed an important point, considering that the 6GTG (Cas12a – I4 Conformation) and 5NFV (FnCas12a bound to R-loop) structures underwent major conformational changes over

~6–7 μs of continuous MD (Figs. 2 and 3). Importantly, the simulations also show that the RuvC active site with the DNA TS is remarkably stable (Supplementary Fig. 9b–d).

We performed Umbrella Sampling (US) simulations[30], studying the traversal of the DNA TS along two reaction coordinates (RCs). Here, the complete Cas12a structure (Fig. 1c) with the full-length TS close to the REC2 domain was the initial state, while the catalytically competent Cas12a structure obtained after ~10-μs-long MD simulation (Supplementary Fig. 8c) was the final state. Structural[9–14], single-molecule[12,16,19,21–24] and computational[25] studies suggested that conformational changes in the Cas12a protein are needed to conduct the traversal of the TS. Hence, the difference in root-mean-square deviation (RMSD) of all the Cα atoms between the initial and the final states appealed as an appropriate RC (RC1) to capture the slowest dynamics involved in the conformational transition. Additionally, the second RC (RC2) was selected as the distance between the center of mass of the RuvC catalytic core and that of the TS region including the scissile phosphate (i.e., the DNA nucleobases at positions −2 to −4, as indicated by biochemical and single-molecule data)[22,27]. The conformational transition was sampled over ~50 μs. Upon removing the biases incurred by the restraints, the free energy landscape (i.e., the Potential of Mean Force, PMF) was described along its minimum free energy pathway (see the Methods section, Supplementary Figs. 10–12).

We observe that the free energy profile (Fig. 4a, blue line) flows freely through a plateau (i.e., state A), barrierless toward the first minimum (state B) as the traversal of the DNA TS is fostered by the interactions of the α-helical lid with the crRNA:TS duplex. Indeed, the α-helical lid anchors the duplex with its positively charged residues (K1013, R1014, R1016, K1018), while interacting with the nucleobases using N1009 (Supplementary Fig. 13a, c–f). A phenylalanine residue (F1012) is also observed to establish stacking interactions with the nucleobases of the TS (with $G_{-1}$ and $T_{-2}$; Supplementary Fig. 13b). These critical interactions with the lid residues guide the descent of the TS. The α-helical lid shows high dynamicity as it traverses through state A (Supplementary Fig. 14a) and restores α helicity again along the minimum free energy path (Supplementary Fig. 14a–c) as the profile slides down to state B and establishes extensive network of interactions with the crRNA:TS duplex (Supplementary Fig. 13). Then, a ~3 kcal/mol barrier is observed leading to the second minimum (state C) corresponding to the DNA TS repositioning within the catalytic core, and the closure of the Nuc domain toward the core of the enzyme (Fig. 4a and Supplementary Fig. 15c, d), constituting the slowest step in the process. These observations suggest that Nuc is critical in aiding the accommodation of the DNA TS within the RuvC catalytic core. Accordingly, the K1206-R1218 α-helix/β-sheet region of Nuc maintains persistent interaction with the 5′-tail of the DNA TS (Supplementary Fig. 15e). Hence, Nuc could hold the TS within the RuvC active core for catalysis, as also supported by recent single-molecule studies[22]. Moreover, mutational studies on the Nuc β-sheet region reported compromised TS cleavage activity[9,11]. Taken together, Nuc is critical in stabilizing the DNA TS at the level of the RuvC catalytic pocket, aiding its accommodation for cleavage, as demonstrated by our long time-scale simulations.

### Mutations of the lid adversely affect target strand traversal

To further verify the role of the lid in the traversal of the TS, we mutated the positively charged residues in the lid (K1013, R1014, R1016, K1018), along with F1012 and N1009, into alanine and we performed free energy simulations. The free energy profile was generated by collecting an ~50 μs ensemble, following the same US approach of the wild-type (WT) system.

The PMF of the mutated system shows an additional minimum at the level of state A with a free energy barrier of ~5 kcal/mol, which impedes a facile traversal of the TS (Fig. 4a, red line) as it was observed in the WT system. This is because, in the mutated system, the TS

locates more closely to the REC2 domain owing to the failure of interactions with the mutated lid (Supplementary Fig. 10c). The PMF of the mutated system also shows a second free energy barrier of ~3 kcal/mol corresponding to the TS repositioning within the RuvC catalytic core and the closure of Nuc, similar to the WT system (Supplementary Fig. 10). This indicates that alanine mutations of the lid mainly affect the traversal of the TS, with a negligible impact on the motions of major domains like REC2 and Nuc (Supplementary Fig. 15a–d). The lid loses its α-helicity briefly and most likely owing to alanine mutations[31], the α helicity is then restored to a stable value (Supplementary Fig. 14a, d) through the length of the minimum free energy path.

The role of the aforementioned lid residues (N1009, F1012, K1013, R1014, R1016 and K1018) in aiding the traversal and accommodation of the TS in the RuvC catalytic core was further investigated by performing alchemical free energy simulations, using thermodynamic integration with softcore potentials (details are reported in the Methods section)[32]. We computed the relative free energy changes (i.e., ΔΔG) of the individual residues with respect to alanine in binding the TS at both the states encompassing the traversal of the TS guided by the α-helical lid, i.e., states A and B (Fig. 4b). In state A, during the traversal of the TS chaperoned by the lid, N1009 and F1012 are the major contributors in binding the TS. K1013 and R1016 also play critical roles by interacting with the backbone of the TS, as also observed during classical simulations (Fig. 2). In state B, when the TS has reached the RuvC catalytic core and is primed toward further readjustments for its accommodation, F1012 stands out as the key player in the process, while N1009, K1013, and R1016 continue to contribute in maintaining the bound state.

To experimentally verify our observations, we introduced point mutations of residues in Cas12a that were identified through molecular simulations to facilitate TS traversal and assayed the TS cleavage activities of the mutant proteins in vitro (Fig. 4c), where the E1006Q variant is the catalytically inactive Cas12a treated as the negative control. We observed significant reductions in TS cleavage with individual substitutions of the positively charged residues such as K1013A and R1016A, while moderate reductions were observed for point mutations of N1009A, R1014A and K1018A (Fig. 4c, Supplementary Fig. 17). This is in accordance with alchemical free energy simulations where the free energies of K1013 and R1016 are significantly higher than R1014 and K1018 at both states A and B (Fig. 4b). A substantial reduction in TS cleavage activity was also observed with combined alanine substitutions of all positively charged residues (K1013A, R1014A, R1016A, K1018A) in the lid, most likely due to an additive impact from K1013A and R1016A. In light of our simulations, this can be attributed to the loss of anchoring interactions between the lid and the crRNA:TS duplex (Fig. 2). Notably, we also observed a significant reduction in TS cleavage with a single phenylalanine-to-alanine substitution (F1012A), as well as when combined with the other positively charged residues in the lid. The phenyl ring of F1012 was observed to stack with the nucleobases during umbrella sampling simulations of the TS traversal (Fig. 4a) and displays a marked energetic contribution in state B (Fig. 4b), suggesting that F1012 plays a critical role in guiding the TS toward the RuvC core. Taken together, our computational simulations and experimental validation establish a pivotal role of the α-helical lid in the traversal of the TS toward the RuvC catalytic pocket.

## Discussion

In the CRISPR-Cas12a genome-editing system, the presence of a single RuvC catalytic site raises the question on how the enzyme could sequentially perform cleavages of the DNA NTS and TS respectively[9,12,19]. Structures of Cas12a show that the TS stays spatially distant from the RuvC catalytic core, close to the REC lobe (Fig. 1). This implies that, following the NTS cleavage, a conformational change would allow the accommodation of the TS within the RuvC catalytic

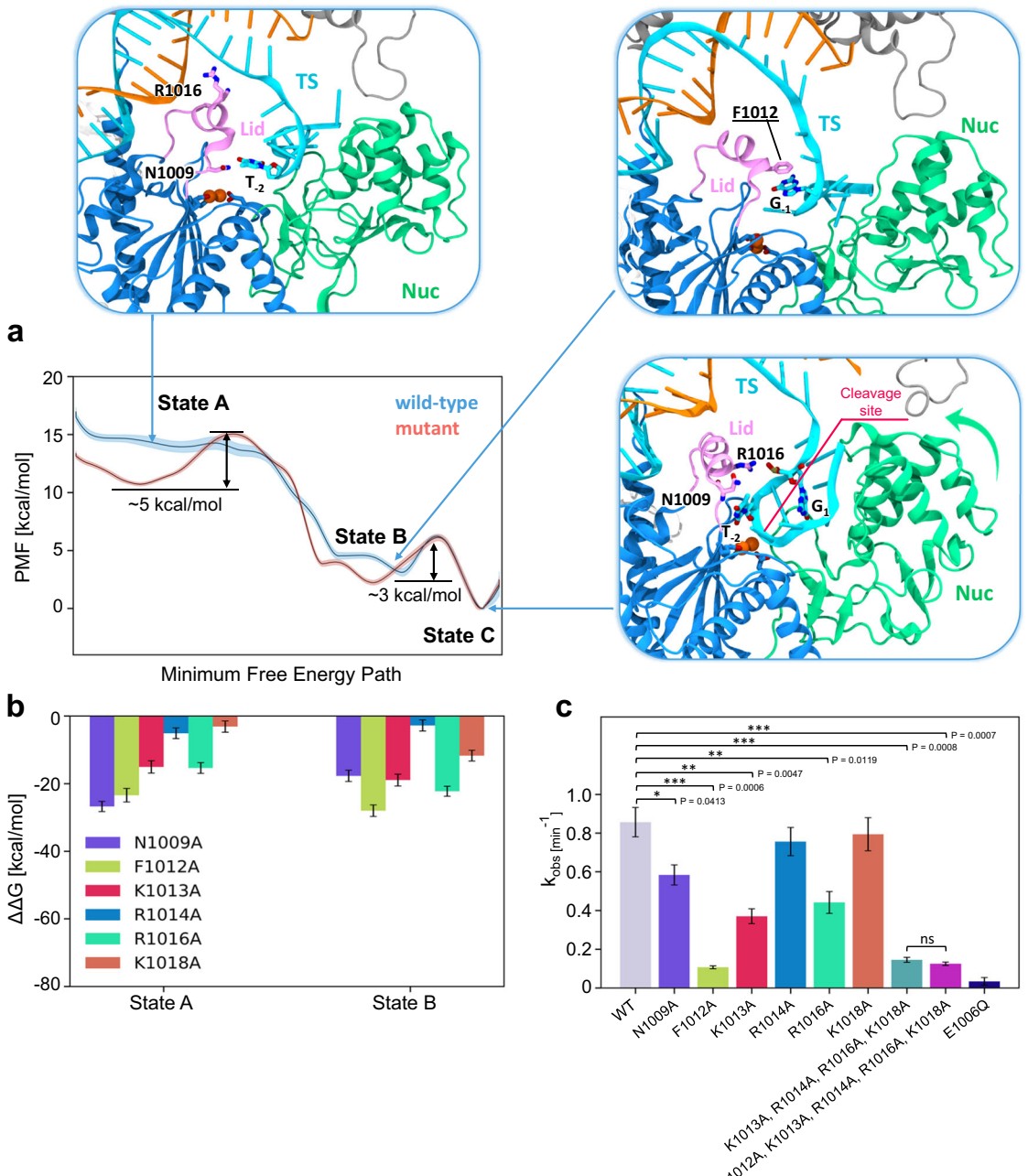

**Fig. 4 | Traversal of the DNA target strand toward the RuvC catalytic core and role of the α-helical lid. a** Free energy profiles for the traversal of the DNA target strand (TS) toward the RuvC core, in the wild-type (WT) Cas12a (blue) and upon mutating relevant residues of the α-helical lid (N1009, F1012, K1013, R1014, R1016, K1018) into alanine (red). The Potential of Mean Force (PMF, in kcal/mol) is computed from 2-D umbrella sampling simulations and plotted along the minimum free energy path (see the Methods section). Representative snapshots, indicated by arrows, are based on the reaction coordinate values along the minimum free energy path of the wild-type Cas12a. Details for the Cas12a mutants are reported in Supplementary Fig. 10. **b** Relative free energy changes (ΔΔG, in kcal/mol) of the individual residues with respect to alanine in binding the DNA TS at both the states

encompassing the traversal of the TS guided by the α-helical lid, i.e., states A and B. Binding free energies are computed through the Multistate Bennett Acceptance Ratio (MBAR) method[60,61] (see the Methods section) and denoted with the associated MBAR errors. **c** Quantitation of in vitro DNA cleavage rates ($k_{obs}$) of Cas12a show F1012A and the combined mutation (K1013A, R1014A, R1016A, and K1018A) display a markedly significant reduction in TS cleavage, while K1013A and R1016A also significantly affect cleavage. Data represents single exponential fitting of $n = 3$ independent replicas and is plotted with the mean ± standard error of mean. Statistical significance was calculated using two-tailed unpaired $t$-test ($P$-value reference: ns $P > 0.05$, *$P \le 0.05$, **$P \le 0.01$, ***$P \le 0.001$).

pocket, but the molecular details of this process have remained elusive.

Here, we performed extensive MD simulations of Cas12a after NTS cleavage, collecting an overall ensemble of ~170 μs. The simulations reveal a mechanism for the traversal of the DNA TS toward the RuvC catalytic core, in which an α-helical lid plays a pivotal role. Structural studies showed that this α-helical lid occludes the RuvC catalytic core

in the binary RNA-bound Cas12a complex and opens for the accommodation of the NTS within the catalytic site[12,33]. Multi-μs long MD simulations of Cas12a after NTS cleavage show that the α-helical lid establishes extensive interactions with the crRNA:TS duplex in its open conformation (Fig. 2a, b). These interactions involve the positively charged residues, including K1013 and R1016, that anchor the PAM-distal end of the crRNA:TS duplex. A bending of the TS at the end of the

duplex is also observed, followed by the TS region including the scissile phosphate moving toward the RuvC site (Fig. 2c, d). This observation is in agreement with experimental findings of a distortion of the DNA TS, downstream of the crRNA:TS duplex, which was indicated to be necessary for it to reach the RuvC cleavage site[27].

The REC2 domain also displays significant flexibility[12,22,25]. Indeed, the REC2b domain, composed of short alpha helices connected by turns, shows prominent inward rearrangement, which occurs in concert with the arching of the terminal major groove of the crRNA:TS duplex (Fig. 3a). This suggests that the large REC2b domain aids in the TS traversal by pushing the PAM-distal major groove of the duplex toward the inner core of the protein. Both conformational changes occur concertedly after the binding of the α-helical lid to the duplex (Figs. 2, 3, and Supplementary Fig. 6). This indicates that the binding of the α-helical lid prompts the arching of the duplex, induced by REC2b pushing inward. Thus, the cooperativity between REC2 and the α-helical lid could foster the traversal of the DNA TS toward the RuvC core. The Nuc domain also bends inward, in concert with the bending of the DNA TS (Fig. 3b). As REC2b and Nuc bend, they also significantly increase their interactions along the long timescale dynamics (Supplementary Fig. 7c), resulting in a closed conformation of the complex. In this respect, cryo-EM and single-molecule experiments revealed high flexibility and coupled dynamics of REC2 and Nuc[12], also suggesting an open-to-close conformational change of the two domains[21,23,24]. Single-molecule experiments further indicated such conformational changes to be crucial for DNA cleavage[12,16,22]. Considering this evidence, the synergistic conformational changes of the major domains observed here, along with the anchoring of the α-helical lid, poise the TS to traverse toward the RuvC active site for cleavage.

To elucidate the mechanism of DNA TS traversal toward the RuvC catalytic core, we performed free energy simulations. We found that the traversal of the TS occurs barrierless toward a first free energy minimum (Fig. 4a). This free-flowing profile is fostered by the interactions of the α-helical lid with the duplex. Indeed, positively charged residues of the lid establish extensive electrostatic interactions with the TS, while F1012 stacks the nucleobases aiding the TS traversal. In this respect, it is interesting to note that, though we found proactive interaction of the lid with the TS in both the ~10-μs-long unbiased runs (Fig. 2) and the umbrella sampling simulations (Fig. 4), yet the classical MD runs could not capture the barrierless transitioning of the TS toward the RuvC active site as seen from state A to B (Fig. 4). This barrierless transition involves an ~18 Å reduction between the $P_{SCI}$ and the RuvC core, whereas classical MD could only capture the initial ~6 Å reduction. Free energy simulations also show that mutations of the residues in the lid interacting with the duplex impede a facile traversal of the TS with a ~5 kcal/mol barrier. Calculation of the relative free energy changes of the individual residues with respect to alanine in binding the TS at critical states of its traversal (Fig. 4b) confirmed the pivotal role of the lid residues, especially spotlighting the contribution of F1012 in binding the TS. Our biochemical experiments also showed that alanine mutations of the interacting residues adversely impact the cleavage of the DNA TS (Fig. 4c). While the combined mutations of the positively charged residues in the lid (K1013A, R1014A, R1016A, K1018A) result in a notable reduction of TS cleavage, single point mutation of F1012 alone plummets TS cleavage manifold times (Supplementary Figs. 17 and 18). K1013A, and R1016A also reduce DNA cleavage, in line with binding free energy simulations. Finally, the relative free energy changes for N1009 drop moving from state A to B, which is reflected by a more moderate reduction of DNA cleavage, compared to F1012A, K1013A, and R1016A. Taken together, the lid interactions observed during our simulations establish a pivotal role of the α-helical lid that guides the TS toward the RuvC core. In line with these observations, truncation of the lid or perturbation of its adjacent helix 1 region have also been shown to hamper target DNA

cleavages[12,34]. Moreover, a structural study on Casφ, a miniature Cas protein phylogenetically related to Cas12a, revealed a similar α-helical lid exposing the RuvC domain for cleavage[35]. Alanine substitution of the lid residues in this Casφ protein abolished double-stranded DNA cleavages, thereby underscoring the importance of the lid in the functioning of the enzyme.

Free-energy simulations also reveal that the Nuc domain is critical in aiding the accommodation of the TS within the RuvC core. Indeed, a ~3 kcal/mol barrier corresponds to the closure of the Nuc domain and further repositioning of the TS within the core of the enzyme (Fig. 4a). This region of Nuc maintains persistent interactions with the 5′-tail of the DNA TS and mutational studies of this region adversely impact TS cleavage[9,11]. This suggests that Nuc could hold the TS at the RuvC site for cleavage, as also hinted by single-molecule experiments[22]. Furthermore, free energy simulations of Cas12a including a mutated lid show that the closure of Nuc remains unaffected, and mutations in the lid rather hamper only the traversal of the DNA TS. Therefore, our findings indicate that while the lid is critical for the traversal of the DNA TS, Nuc can be decisive in the accommodation and repositioning of the TS in the RuvC catalytic core.

Taking together our findings and the available cryo-EM, biochemical and single-molecule experimental data, we propose a model for the traversal of the DNA TS strand toward the RuvC core for cleavage (Fig. 5). In this model, the α-helical lid holds a critical role, along with the conformational changes in the major domains. Upon NTS cleavage, the α-helical lid remains in its open conformation and anchors the crRNA:TS duplex, establishing electrostatic interactions (Fig. 5a, b). Then, the REC2 domain pushes the terminal major groove of the duplex through its REC2b region moving inward (Fig. 5c). As REC2b pushes the crRNA:TS duplex, the Nuc domain simultaneously bends toward the core of the enzyme, aiding the bending of the DNA TS. These cooperative motions of the major domains prime the DNA TS for its traversal. The α-helical lid, now being in close proximity to the TS region around the scissile phosphate, establishes elaborate interactions with the TS and guides it toward the RuvC catalytic core. Finally, the closure of Nuc aids the TS to reposition and accommodate within the active site (Fig. 5d).

## Methods
### Structural models
Molecular simulations were based on two structures of the *F. novicida* Cas12a obtained upon NTS cleavage: (1) the cryo-EM structure EMD-0065 (PDB: 6GTG (Cas12a – I4 Conformation))[12] at 3.27 Å resolution and (2) the X-ray structure PDB: 5NFV (FnCas12a bound to R-loop)[9], at 2.50 Å resolution. The 6GTG (Cas12a – I4 Conformation) structure contains an α-helical "lid" (residues L1008 – K1021, Fig. 1a), which is structurally disordered in the 5NFV (FnCas12a bound to R-loop) structure. Hence, missing residues in the 5NFV (FnCas12a bound to R-loop) structure were reconstructed through homology modeling using the SWISS-MODEL software (Fig. 1b)[36]. A third and more complete model system of Cas12a after NTS cleavage (Fig. 1c) was built starting from the 6GTG (Cas12a – I4 Conformation) structure. Since this structure lacks the region of the TS downstream of the crRNA:TS duplex, we modeled this region of the DNA based on the 5NFV (FnCas12a bound to R-loop) X-ray structure. The catalytically competent state of Cas12a for TS cleavage was also considered, based on related crystal structures holding the DNA TS within the RuvC active site (see Supplementary Methods). All systems were embedded in explicit waters and counterions were added to neutralize the total charge, leading to periodic cells of ~138*149*167 Å³ and ~307,000 atoms for each system.

MD simulations were performed using the Amber ff19SB[37] force field, with the ff99bsc1 corrections for DNA[38] and the ff99bsc0 + χOL3 corrections for RNA[39,40]. The TIP3P model was employed for water[41], and the Li & Merz 12-6 model of non-bonded interactions was used for

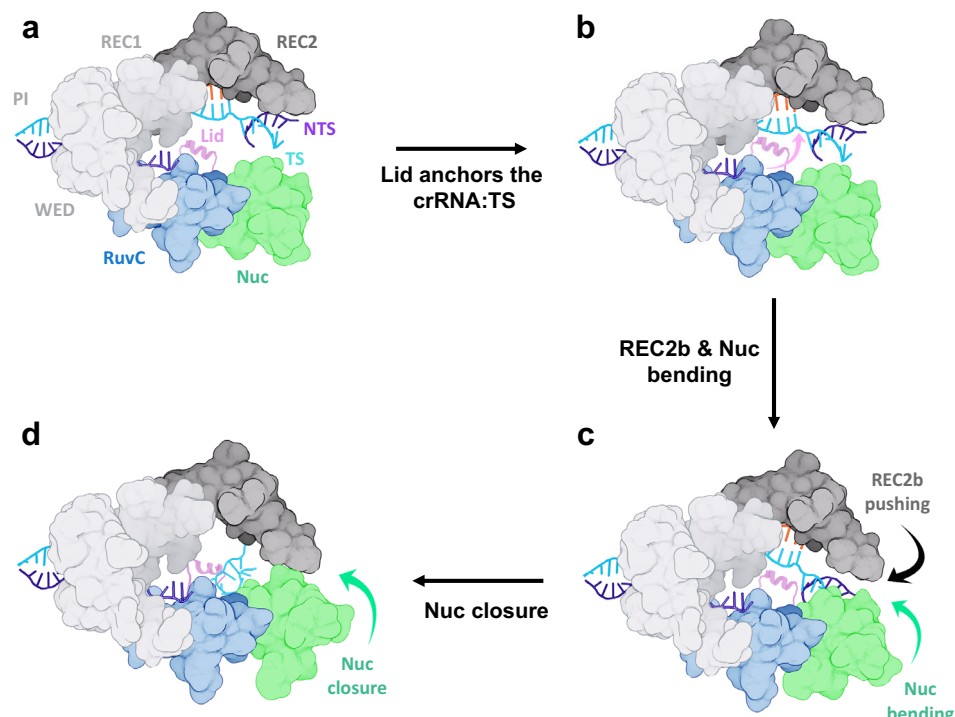

**Fig. 5 | Model for the DNA target strand traversal toward the RuvC catalytic core in CRISPR-Cas12a. a** Upon cleavage of the DNA non-target strand (NTS), the α-helical lid assumes an open conformation, exposing the RuvC catalytic core, while the DNA target strand (TS) locates close to the REC2 domain[12]. **b** The α-helical lid, in its open conformation, anchors the crRNA:TS duplex by establishing electrostatic interactions. **c** The inward bending of the REC2 and Nuc domains pushes the crRNA:TS duplex, while aiding the bending of the DNA TS and facilitating its conformational transition. **d** The α-helical lid binds the DNA TS and guides it toward the RuvC catalytic core. The closure of Nuc aids the TS to accommodate within RuvC for cleavage.

$Mg^{2+}$ ions[42]. We have extensively employed these force field models in computational studies of CRISPR-Cas systems[43], showing that they perform well for long timescale simulations on Anton-2[44]. The Li & Merz model also reported a good description of $Mg^{2+}$ bound sites, in agreement with quantum/classical simulations[45]. An integration time step of 2 fs was employed. All bond lengths involving hydrogen atoms were constrained using the SHAKE algorithm. Temperature control (300 K) was performed via Langevin dynamics[46], with a collision frequency γ = 1. Pressure control was accomplished by coupling the system to a Berendsen barostat[47] at a reference pressure of 1 atm and with a relaxation time of 2 ps. The systems were subjected to energy minimization, thermalization, and equilibration as detailed in the Supplementary Methods. Then, ~120 ns of conventional MD were carried out in an NVT ensemble using the GPU-empowered version of AMBER 20[48]. The well-equilibrated systems were used as starting points for simulations on Anton-2[26], a special-purpose supercomputer for multi-μs long MD simulations.

Long timescale MD simulations on Anton-2 were performed using the same force field parameters used for conventional MD simulations. A reversible multiple time step algorithm[49] was employed to integrate the equations of motion with a time step of 2 fs for short-range nonbonded and bonded forces and 6 fs for the long-range nonbonded forces. Simulations were performed in the NPT ensemble using the multigrator integrator as implemented in Anton-2[50]. Pressure control was accomplished via the Martyna, Tobias, Klein (MTK) barostat[51], set to maintain 1 bar of pressure, with a tau (piston time constant) parameter of 0.0416667 ps and reference temperature of 310.15 K. The barostat period was set to the default value of 480 ps per timestep. Temperature control was accomplished via the Nosé-Hoover thermostat[52,53] with a tau (time constant) parameter of 0.0416667 ps. The k-Gaussian split Ewald method[54] was used for long-range electrostatic interactions. Hydrogen atoms were added assuming standard

bond lengths and were constrained to their equilibrium position with the SHAKE algorithm[55]. By using this approach, each system was simulated on Anton-2, obtaining continuous MD runs of ~10 μs and in replicates. We obtained multiple trajectories of the 6GTG (Cas12a – I4 Conformation) cryo-EM structure (simulated for ~10 μs) and its more complete model including a longer DNA (~10 μs in two replicates), as well as of the 5NFV (FnCas12a bound to R-loop) X-ray structure (~10 μs in two replicates). The structural model of the catalytically competent Cas12a including the DNA TS within the RuvC site was also simulated for ~10 μs. This resulted in a total of ~60 μs of all-atom MD simulations.

**Umbrella Sampling simulations**

The umbrella sampling (US) method[30] was used to compute the free energy profile associated with the traversal of the DNA TS toward the RuvC active site. In this method, a number of simulations (US windows) are run in parallel with additional harmonic bias potential applied to selected Reaction Coordinates (RCs):

$$V(RC) = \frac{k}{2}(RC(t) - RC^*)^2 \qquad (1)$$

where $V(RC)$ is the value of the bias potential, $k$ is a bias force constant, $RC(t)$ is the value of $RC$ at given time $t$ and $RC^*$ is the reference value of $RC$. By using different $RC^*$ values in each US window, one can sample the biased probability distribution $p_b(RC)$ along the whole $RC$ range of interest. We performed two-dimensional (2-D) US simulations using two $RCs$. $RC1$ was the difference in root-mean-square deviation (RMSD) of the Cα atoms between the initial (Fig. 1c) and the final (Supplementary Fig. 8c) states. $RC2$ was the distance between the center of mass of the RuvC catalytic core (D917, E1006, D1255) and the TS region including the scissile phosphate (i.e., the DNA nucleobases at positions −2 to −4, as indicated by biochemical and single-molecule

experiments)[22,27]. $RC1$ was divided into bin sizes of 0.2 Å $RMSD$ difference and each window was run for ~10 ns with a collective force constant $k$ of 50 kcal mol$^{-1}$ Å$^{-2}$. $RC2$ was discretized into bin sizes of 0.2 Å and individual windows were run for ~15 ns with a collective force constant $k$ was 50 kcal mol$^{-1}$ Å$^{-2}$. Overall, we simulated 28 ($RC1$) * 119 ($RC2$) = 3332 windows, resulting in a collective ensemble of ~50 μs simulations.

Two independent sets of 2-D US simulations were performed at 300 K (each for ~50 μs): (i) for the wild-type system and (ii) upon mutating relevant residues of the α-helical lid (N1009, F1012, K1013, R1014, R1016, K1018) into alanine. Then, the free energy profiles were computed using the Weighted Histogram Analysis (WHAM) method[56] upon removing the initial 1/3rd of the US trajectories from each window, corresponding to the relaxation of the system. Analysis of the conformational ensembles was performed on the reweighted trajectories (see Supplementary Methods). Analysis of the free energy landscape was performed by computing the minimum free energy pathway using an approach similar to the Dijkstra algorithm[57]. In this approach, the map is sampled from the origin point, propagating to the lowest energy point available on each iteration until it reaches the target point. This enabled us to consider both the $RCs$ along a minimum free energy path. The error estimation on the minimum free energy pathways was performed using the Monte Carlo bootstrap error analysis[58]. Full details on the 2-D US simulations are reported in the Supplementary Methods, including the 2-D free energy surfaces (Supplementary Fig. 11) and the convergence of the free energy profiles (Supplementary Fig. 12).

### Alchemical free energy perturbation

We computed the relative free energy changes (i.e., $\Delta\Delta G$) of the individual residues with respect to alanine in binding the DNA target strand (TS), by performing alchemical free energy simulations[59]. Here, the DNA target strand (TS, ligand) bound to the Cas12a complex (receptor) was decoupled to an unbound state following the alchemical free energy formalism using softcore (SC) potentials[32], by gradually diminishing the electrostatic and van der Waals interactions of the TS. We used a lambda (λ) dynamic variable to define the thermodynamic states of the system along this decoupling alchemical pathway. Thermodynamic integration (TI) simulations were performed at 21 λ values from 0 to 1, at an interval of 0.05, for the wild-type (WT) CRISPR-Cas12a and for each individual alanine mutant. The systems were minimized and equilibrated using the simulation protocol described above. The final binding free energies were computed using the Multistate Bennett Acceptance Ratio (MBAR) method[60,61] to integrate the free energies over the different lambda values[62,63]. The relative binding free energy ($\Delta\Delta G$) was calculated as follows:

$$\Delta\Delta G_{residue} = \Delta G_{WT} - \Delta G_{mutated} \qquad (2)$$

For each system (i.e., the WT CRISPR-Cas12a, and the N1009A, F1012A, K1013A, R1014A, R1016A, K1018A mutants), we considered two states of the CRISPR-Cas12a complex during the traversal of the DNA TS (states A and B, Fig. 4). Alchemical simulations were performed at 21 values of λ (from 0 to 1, in steps of 0.05), reaching ~30 ns for each window. Convergence of the $\Delta\Delta G$ for each system is reported in Supplementary Fig. 16. Considering all the systems and states investigated here, a total of ~8.8 μs sampling was reached (full details are in the Supplementary Methods).

### FnCas12a expression and purification

The DNA sequence of *Francisella tularensis* subsp. novicida U112 (Fn) Cas12a (WP_003040289) was codon optimized for heterologous expression in Escherichia coli (*E. coli*) and synthesized by GeneArt (Thermo Fisher Scientific). The FnCas12a gene was inserted into the 1B plasmid (Addgene #29653) using ligation-independent cloning (LIC),

resulting in a construct carrying an N-terminal hexahistidine tag followed by a tobacco etch virus (TEV) protease cleavage site. Point mutations were introduced by Gibson assembly of the PCR-amplified vector backbone with gBlock Gene Fragments (IDT) encoding the individual mutations. The sequences of the synthetic gene, primer and gBlocks are listed in Supplementary Data 1. Mutant FnCas12a constructs were purified as for wild type. Purification of Cas12a was done as described[64,65]. In brief, Cas12a constructs were expressed in *E. coli* BL21 Rosetta2 (DE3) cells (Novagen, Wisconsin, USA). Cells were lysed in 20 mM Tris pH 8.0, 500 mM NaCl, 5 mM imidazole, 1 μg/mL pepstatin, 200 μg/mL 4-(2-Aminoethyl)benzenesulfonyl fluoride hydrochloride (AEBSF) by ultrasonication. Clarified lysate was applied to a 10 ml Ni-NTA (Sigma-Aldrich) affinity column. The column was washed with 20 mM Tris pH 8.0, 500 mM NaCl, 5 mM imidazole, and bound protein was eluted by increasing imidazole concentration to 250 mM. Eluted protein was dialyzed against 20 mM HEPES pH 7.5, 250 mM KCl, 1 mM DTT, 1 mM EDTA overnight at 4 °C in the presence of TEV protease to remove the 6xHis- affinity tag. Cleaved protein was further purified using a HiTrap HP Heparin column (GE Healthcare, Illinois, USA), eluting with a linear gradient to 1.0 M KCl. Elution fractions were pooled, concentrated, and further purified by size exclusion chromatography using a Superdex 200 (16/600) column (GE Healthcare) in 20 mM HEPES-KOH pH 7.5, 500 mM KCl, 1 mM DTT yielding pure, monodisperse proteins. Aliquots were flash-frozen in liquid nitrogen and stored at −80 °C.

### FnCas12a nuclease activity assays

In vitro nuclease activity assays were conducted using purified WT or mutant FnCas12a proteins programmed with a crRNA targeting the ⌐-sequence (oMS017, IDT, HPLC purified) and a dsDNA substrate containing fluorescently labeled TS (oDS285:oDS271, Merck, HPLC purified). FnCas12a and crRNA were mixed in a ratio of 1:1.2 and incubated for 10 min at 25 °C to allow binary complex formation. Reactions were started by addition of target dsDNA (FnCas12a:dsDNA, 10:1) and incubated at 37 °C. All samples were assembled in a final reaction volume of 20 μL containing 0.5 μM (mutant) FnCas12a, 0.6 μM crRNA and 50 nM dsDNA in a final buffer of 10 mM HEPES-KOH pH 7.5, 250 mM KCl, 5 mM MgCl$_2$, 0.5 mM DTT. Reactions were stopped at indicated time points by addition of EDTA and Proteinase K (Thermo Fisher Scientific) in final concentrations of 80 mM and 0.8 mg/mL, respectively and incubated for 15 min at 37 °C. Samples were mixed with equal volume of a 2X dPAGE loading dye (95% formamide, 25 mM EDTA), heated to 95 °C for 5 min and resolved on a 15% denaturing (7 M Urea) polyacrylamide gels run in 0.5 X TBE buffer. Assays were conducted in technical triplicates; indicated error bars represent the standard error of mean. Fluorescence of the ATTO532-labeled substrate and cleavage products was detected using a Typhoon FLA 9500 gel imager; the cleavage rate was quantified based on loss of uncleaved substrate DNA using ImageQuant TL v.8.2.0. Sequences of RNA and DNA oligos utilized in the nuclease assays are listed in Supplementary Table 3.

### Statistics and reproducibility

All simulations were performed in replicates, as also specified in the main text and pertinent figure legends. Quantitation data of in vitro DNA cleavage assay represents single exponential fitting of $n = 3$ independent replicas and is plotted with the mean ± standard error of mean. Statistical significance was calculated using two-tailed unpaired $t$-test.

### Ethics and inclusion statement

Molecular simulations and analyses were performed at the University of California, Riverside (United States), primarily by international graduate students and post-doctoral scholars. Experimental data were generated at the University of Zürich (Switzerland). We carefully

evaluated and mutually agreed upon researcher contributions and authorship criteria, ensuring equity in research collaborations.

## Reporting summary

Further information on research design is available in the Nature Portfolio Reporting Summary linked to this article.

## Data availability

All data are included in the article and Supplementary Information. Molecular simulations were based on two structures of the *F. novicida* Cas12a obtained upon NTS cleavage: (1) the cryo-EM structure EMD-0065 (PDB: 6GTG (Cas12a – I4 Conformation)) solved at 3.27 Å resolution and (2) the X-ray structure PDB: 5NFV (FnCas12a bound to R-loop), solved at 2.50 Å resolution. Sequences of oligos, synthetic genes and gene fragments are provided in Supplementary Data 1. In vitro DNA cleavage assay time points data of Cas12a wild type and its mutants is provided in Supplementary Data 2 and the corresponding uncropped gel images are in the Source Data File. Atomic coordinates of the initial and final structure of CRISPR-Cas12a along the traversal of the target strand toward the RuvC active site are available in figshare with the identifier 10.6084/m9.figshare.24978669. More details can be provided by the corresponding author upon request. Source data are provided with this paper.

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

## Acknowledgements

We thank Chinmai Pindi for useful discussions. This material is based upon work supported by the National Institutes of Health (Grant No. R01GM141329, to G.P.) and the National Science Foundation (Grant No. CHE- 2144823, to G.P.). M.J. acknowledges support from the Swiss National Science Foundation (project grant 31003A_182567) and the Howard Hughes Medical Institute (project grant 55008735). Anton 2 computer time was provided by the Pittsburgh Supercomputing Center (PSC) through Grant R01GM116961 from the National Institutes of Health. The Anton 2 machine at PSC was generously made available by D.E. Shaw Research. This work also used Expanse at the San Diego Supercomputing Center through allocation MCB160059 from the Advanced Cyberinfrastructure Coordination Ecosystem: Services & Support (ACCESS) program, which is supported by National Science Foundation grants #2138259, #2138286, #2138307, #2137603, and #2138296. Computer time was also provided by the National Energy Research Scientific Computing Center (NERSC) under Grant No M3807.

## Author contributions

A.S., M.A. and P.R.A. performed molecular simulations and analyzed data. †M.A. and P.R.A. contributed equally. M.S. and C.C. performed DNA cleavage experiments, supervised by M.J. G.P. conceived this research. A.S. and G.P. wrote the manuscript with critical input from all authors.

## Competing interests

The authors declare no competing interests.
