## [Peer Review File · Nature Communications]

An Alpha-helical Lid Guides the Target DNA toward Catalysis in CRISPR-Cas12aReviewer #1 (Remarks to the Author):

The manuscript "An Alpha-helical Lid Guides the Target DNA toward Catalysis in CRISPR-Cas12a" by Saha et al. describes the use of molecular modelling, unbiased and biased molecular dynamics simulations and alchemical free energy simulations to probe the mechanism by which CRISPR-Cas12a attracts the DNA target strand to its active site.

The data presented in this manuscript help to elucidate how Cas12a cleaves the TS after cleaving the NTS, helping to build our understanding of the molecular mechanism of Cas12a, which is crucial for the future development of Cas12-based genome editing tools. I therefore expect this work to be of significance to the CRISPR-Cas field.

The data presented for the most part support the conclusions – I have made suggestions for where the analysis and interpretation might be improved and where additional justification for the conclusions is required.

The computational aspects of this work (I am not qualified to review the experimental part), which comprise the bulk of the results, are largely well described and use established, sound methods. The length of the simulations is above what might often be possible due to use of the Anton2 supercomputer. The methods for the most part were highly comprehensive, something I really appreciate – I've noted the few instances where more information would be helpful.

Overall, the work has been done to a high standard, the manuscript is well written, and the results mostly support the conclusions, which provide new insight into the function of a popular gene editing tool that will aid its future use and engineering. However, I think it would benefit from some minor changes and potentially some additional analyses (or at least alternative presentation of the data) prior to publication – these are outlined below.

I have sorted my comments below into themes as I found that similar issues tended to appear throughout the manuscript. Within these themes, comments are in roughly the same order as the main text. Page numbers refer to the main text other than when it is indicated that they refer to the Supplementary Methods (SM).

Clarification required

Supp Fig 1 shows the distance between selected residues from the α -helical lid (R1014, K1013 and K1018) and either the crRNA or the TS. More information is required about how these distances were calculated – for instance, is it the minimum distance to any atom of the respective nucleic acid strand? Or the distance to the COM of the nucleic acid strand? This information is required to know how to interpret the distances. The generic nature of these distances is in contrast to those plotted in Fig 2, which are between K1013 to C2 of the TS and R1016 to U12 of the crRNA, and Supp Fig 3, where they are between K1013 and G1 of the TS and R1016 and U11 of the crRNA. Why calculate specific distances in some cases and generic distances in others? On what basis were the specific residue/base pairs selected?

Fig 2, P6: It is noted more than once that "biochemical and single-molecule experiments have shown that the cleavage site on the TS can mainly be located between base positions -2 and -4." (see also P10), yet only the distance between the catalytic core and the phosphate connecting the T-2 and G-3 nucleobases is calculated. Why not calculate the distance to the G-3 and A-4 phosphate too?

Fig 2, Supp Fig 1, 3, 4, 5, 7: The time-series data presented throughout this manuscript are shown as a darker line over a lighter line of the same colour. Presumably the darker line is a running average and the lighter line the raw data, but this needs to be stated

somewhere (along with the averaging time period).

In the interpretation of the time-series data presented in Fig 2 and Supp Fig 1 and 4, particular time points are noted as being when transitions occur (sometimes a dashed line is drawn, other times not, even when it is the same data presented, e.g. Fig 2c vs Supp Fig 4d). How were these time points identified? Simply by eye or through some sort of statistical analysis? The 6 μ s time point at which stable salt bridges apparently form in Supp Fig 1b is clearly incorrect – the distances do not become short enough to represent salt bridge formation, nor stable, until close to 7 μ s. The definition of “stable formation” of an interaction is also not provided. It is claimed in the caption of Supp Fig 1 that K1013 and R1014 remain linked to positions 8 and 12 of the crRNA backbone from 2 μ s onwards, yet the distances clearly move beyond 5 Å, which would indicate they are not bound (although this depends on how the distance was calculated, see comments on Supp Fig 1 above) during the remainder of the simulation. Again, in the interpretation of Fig 3a and Supp Fig 4a, it is claimed that the decrease in the bending angle of REC2b “becomes more substantial after \sim 6 μ s”, but there are other times when large changes occur, e.g. just after 3 μ s. Why focus on this one? Additionally, the time-series data of the two replicas split at this time point, with only one replica becoming more bent, but this is not mentioned in the text.

Supplementary Methods, “Calculation of bending angles”: this section would benefit from reference to the various figures with schematics of these angles. However, even the schematics do not appear to represent what is described in the methods, or at least it is hard to relate the schematics to the methods, as the nucleotide bases are not numbered and the planes are not shown. A better schematic should be provided. Moreover, according to the method for constructing the vectors, the time-series should always start at 90°, yet many of the time-series clearly do not start here (see e.g. Supp Fig 7a), so what was the reference structure? It needs to be clearly stated (and justified) in each case.

Fig 2: Part (d) shows the distance between the scissile phosphate and the centre of mass of the RuvC catalytic core. Are the magnesium ions included in the COM calculation? This isn’t clear from the text but the accompanying image (bottom left) would suggest they are.

Supp Fig 2: What is the RMSD of the lid residues computed with respect to? Is this an all-atom or Ca only RMSD?

Supp Fig 7: Part (b) is interpreted in the caption as showing that “the nucleobases containing the scissile phosphate, T-2 (RMSD < 1.5 Å) and G-3 (RMSD < 0.5 Å), report a remarkable stability.”, yet the RMSD for G-1 is very similar to that of T-2 and the RMSD of C-5 is lower than G-1 and T-2, so why is T-2 considered “remarkably stable”?

Supp Fig 7: What is the reference structure for the RMSD time-series shown in part (c)? It cannot be the initial structure as the time-series to not start from 0, yet the initial structure is what it should be for an interpretation in terms of stability during the simulation.

What data support the statement “Accordingly, the K1206-R1218 α -helix/ β -sheet region of Nuc maintains persistent interaction with the 5’-tail of the DNA TS.”? There needs to be a reference to one or more figures.

P18/SM P2: How were the missing residues in 5NFV constructed? Was a template used? Or structural constraints? Why did the reconstruction not produce a helical structure for the lid region?

SM P2: Why wasn’t the X-ray structure of CRISPR-Cas12b (5U30) able to show the active site and Mg²⁺ ions (for which an additional X-ray structure of CRISPR-Cas12i2 (6LTU) was required)?

SM P2: How exactly was the structure of the catalytically-competent state of Cas12a built? What software was used? How were the different structural features combined?

P18/SM P3: The Berendsen barostat does not produce a thermodynamically correct NpT ensemble. Was this barostat only used for the equilibration stages or also used during the production simulation?

P20/SM P6: Were position restraints used in the alchemical free energy perturbation to prevent movement of the TS as its interactions were reduced?

SM P4: In the calculation of RC1, was the C α RMSD calculated over all amino acids in Cas12a, or only some?

Additional analysis required

P8/9: The claims of coupled motion of the REC2 and Nuc domains is in my opinion not fully justified. The presentation of the time-series data in Fig 3 is very difficult to interpret, with the diagonal offset of the time-series making it particularly hard to check the claimed correlation of the time-scales/patterns over time of the paired datasets ("similar trend", "in concert with"). Why not present these data in regular time-series plots like in Fig 2 and many of the supplementary figures (including Supp Fig 5, referred to in this section)? Additionally, to better validate the claims of processes happening in concert, scatter-plots could be used and the correlation between the two properties could be calculated. Another possibility would be to use principle component analysis to identify the "large-scale coupled motions" supposedly exhibited by the major domains.

Additional discussion required

P5: The interactions between Cas12 and the TS differ between the cryoEM and X-ray crystal structures. Can the authors please discuss these differences (which are presumably due to the different lengths of TS in each)?

P6: "Interestingly, this unstructured lid forms salt-bridges with the crRNA and intermittent interactions with the TS (Supplementary Fig. 3a-b). This, in turn, affects the bending of the TS and the movement of the scissile phosphate toward the RuvC catalytic site (Supplementary Fig. 3c-d)." "affect" in what way? Supp Fig 3 does not show any clear trend in the TS bend angle, which remains > 90°, i.e., not really bent (cf angles around 40° that occur with the helical lid). On the other hand, the distance to the catalytic core seems to decrease faster for the unstructured lid compared to the helical lid and reaches similarly short distances. How does this happen without TS bending? These differences mean that the next sentence, "This indicates that the interactions of the lid with both the crRNA and TS are critical to achieving the bent conformation of the TS." is hard to justify, as interactions occur with either lid structure but TS bending only occurs with the helical lid.

P12: The interactions between Cas12a and the TS observed in the unbiased and umbrella sampling simulations are briefly compared; I would like to see a larger discussion of the differences, especially given the apparently barrierless PMF between the state simulated in the unbiased simulations and State B observed in the umbrella sampling – can the authors comment on why this state did not occur in the unbiased simulations?

P12/Fig 4: More discussion is required of the fact that the $\Delta\Delta G$ value for the N1009A mutation is large yet its effect on activity is relatively small.

Labelling and presentation of figures

Fig 1: In part (a) of the figure, the domains are labelled but the REC and NUC lobes referred to in the text on P3 are not – which domains do they each contain? The PI domain labelled in the figure is not defined or mentioned anywhere in the manuscript. Lastly, Rec1 and Rec2 are not fully capitalised here but are in all subsequent figures.

Supp Fig 1: The residues shown in the snapshots accompanying part (a) need to be labelled like they are for the snapshot accompanying (b). Also please check that it is K1013 and not K1018 shown in these snapshots (hard to tell based on the viewing angle).

Supp Fig 1: “R1014 and K1018 approach the DNA TS at positions 2 and 3 respectively” – isn’t it the opposite, i.e., R1014 approaches TS position 3 and K1018 approaches TS position 2? (at least that’s what it looks like from the snapshot accompanying part (b)).

Arrows are used in Fig 3 (and other figures) to indicate the bending of the REC2b and Nuc domains. Similar arrows should be added for the hybrid arching and TS bending, which are even harder to identify in the snapshots.

Supp Fig 5: Part (d) does not add anything that cannot already be seen, more quantitatively, in parts (a-c). If you must keep it, it should be the same width as parts (a-c) so that comparisons between the panels can be easily made.

Supp Fig 8: Please label the various residues and nucleobases drawn in stick format.

Fig 4: Please label the amino acid residues shown in the State C snapshots (and ideally also the nucleotides in all three snapshots).

Fig 5: The REC domains could be more comprehensively labelled.

Typographical errors and unclear phrasing

P5: “Indeed, the α -helical lid draws the TS toward itself by...” – this phrasing is confusing – is the TS being drawn toward itself (as written), or toward the α -helical lid (what I think is meant)?

Fig 2 caption: “through the G-1, G1, C2, C3 and T4” – no “the” is required.

Supp Fig 5: “A contact is considered when the distance between two heavy atoms among the regions of interest is lesser than 3.5 Å.” – should just be “less”.

P14: What is meant by describing the RNA-bound Cas12a complex as “binary”?

Fig 5: “establishing stable salt-bridge interactions: - should be “salt-bridge”

Supp Methods P4: “Restrain forces” \diamond should be “Restraint forces”

Supp Methods P5: “with a collective force constant k was 50” – I think “was” should be “of”?

Reviewer #2 (Remarks to the Author):

In the present manuscript, the authors unravel the molecular basis of the target strand DNA traversal toward the catalytically competent CRISPR-Cas12a state. To accomplish this goal, they combined extensive molecular dynamics (MD) simulations, enhanced sampling techniques, free-energy calculations, and experimental assays.

In contrast to the broadly characterized and studied Cas9 system, Cas12a cleaves the two DNA strands using a single catalytic domain, first, the non-target strand (NTS) and, second, the target strand (TS). This implies that a significant conformational change involving different elements (Cas12a Nuc and Rec2 domains, DNA, and RNA) is required to accommodate the target strand DNA in the RuvC catalytic core once the non-target strand is cleaved. The molecular mechanism through which the target strand traverses toward the catalytic domain upon NTS cleavage was still unknown. Understanding this mechanism at the molecular level is particularly challenging because of the interplay between different dynamic and complex biomolecules and the existence of key transient states that are difficult to capture with experimental techniques. Here, the authors completely reconstruct the DNA traversal mechanism after non-target strand cleavage, identifying key intermediate states, cooperative motions, and molecular interactions that drive the whole process. In particular, an alpha-helical lid is identified as the key structural element that guides the transition toward the catalytic core. The results obtained in the present work bridge the gap with previous experimental observations and harbor relevant information for understanding Cas12a function and engineering.

Briefly, from my point of view the manuscript is well-written and the results obtained in the manuscript are significantly relevant and interesting. The molecular insights provided using computational simulations on the complex DNA traverse mechanism in Cas12a pave the way toward the application of this methodology to further understand and engineer genome-editing in Cas12a. In particular, the strong points of the present manuscript are:

- Long-time scale simulations. The extensive microsecond MD simulations performed in this work offer an unprecedented detail of key features of the DNA traversal toward the catalytically competent state. Previous MD simulations (up to one microsecond) could not capture most of the key motions described in the present work, which occur after a few microseconds of simulation time. With the 10 microsecond simulations starting from the 6GTG conformation with longer TS (after non-target strand cleavage) it has been possible to identify some of the key features of the process including the recognition of crRNA:TS duplex by alpha-helical lid, the bending of TS and the approach toward the RuvC core, which occur together with cooperative motions of REC2 and Nuc domains.

- Robust computational protocol. The computational protocol used to characterize the complete traversal mechanism is robust. Starting from extensive sampling of the initial and final states to identify key interactions/features followed by a definition of two reaction coordinates for umbrella sampling to reconstruct the complete mechanism. From these simulations, the minimum free energy path is analyzed to explain, step by step, the DNA traversal mechanism for the wild-type and generated mutant.

- Proposal of a model mechanism for the DNA target strand traversal. The MD-based simulations reveal the evolution of the system from the non-catalytic state to the catalytically competent state. Identifying a series of intermediate states that were hidden for experimental techniques. In particular, the simulations reveal the role of the alpha-helical lid guiding the DNA TS toward the catalytic RuvC core.

- Molecular basis. From the MD-based simulations key interactions between the helical lid and DNA TS have been identified for the key intermediate states of the process. It is also really interesting the role of F1012. Mutations based on computational predictions were validated with experimental assays confirming the hypothesis that the helical lid plays a critical role in the first steps of the traversal process.

I have some comments/suggestions regarding the results presented in the present work that will require providing additional details.

1. In PDB 5NFV and in the MD simulations based on this initial structure, the lid is unstructured and flexible. However, the structured alpha-helical lid is selected when

building the catalytically competent Cas12a (Supplementary Figure 6) and in PDB 6GTG with the large TS (Figure 2), which are also the final and initial states of the umbrella sampling simulations, respectively. Therefore, the question is: which is the role of the disordered lid in the DNA traversal mechanism? or in other words, is the disordered lid functionally relevant? The authors should clarify this aspect in the manuscript.

2. Related to the previous question, is the alpha-helical lid changing its conformation/order when going from state A to C in the free-energy simulations? Besides the initial approach toward the crRNA:TS, in Figure 4 it looks like there are some structural changes during the process (which are more pronounced in the mutant, see state A in Supplementary Figure 8 which is partially unstructured). For example, is F1012 always pointing toward TS DNA or it depends on the order/disorder of the lid? It would be interesting to see if the helical character is particularly stabilized in a particular state along the traversal process.

3. In the first and second section of the results (i.e. helical lid anchor crRNA:TS and Rec2 and Nuc motions), a lot of emphasis is put on the some key distances and angles. However, these key features are not analyzed in the free energy simulations. I think that it would be interesting to analyze how these features/interactions change (if possible) along the minimum free-energy path (e.g. which interactions between the alpha-helical lid and TS are formed and broken or the process of TS bending). This will provide additional molecular information on the sequence of events depicted in Figures 4 and 5.

4. One of the major findings is the cooperative motions between different structural elements that facilitate TS traversal (helical lid, Rec2, and Nuc). Based on these results and the scheme shown in Figure 5, first the alpha-helical lid approaches TS DNA and, then, the coupled motion of Nuc and Rec2 occurs. In a previous work (J. Chem. Inf. Model. 2020, 60, 12, 6427–6437), the authors explored the essential dynamics (using principal component analysis) of Cas12a in their DNA and RNA bound states. I wonder if performing a similar analysis for the MD simulations of 6GTG with the large TS may provide relevant information about the coupled motions of the three domains: helical lid, Rec2 and Nuc. Is the helical lid that triggers Rec2 and Nuc closure or all motions are in some way correlated? This should only be added in the Supplementary Material if it provides additional information on the potential cooperativity.

5. How are the representative snapshots of States A, B, and C (shown in Figure 4 and also the starting point for TI simulations) obtained? Are they obtained based on visual inspection/manual selection or using some clustering algorithm? This should be specified in the Supplementary Methods.

6. In Figure 4c, the results of the in vitro DNA cleavage for the mutation E1006Q are shown but not discussed in the manuscript.

Additionally, I have some minor comments/suggestions about some aspects that need to be clarified:

7. In some cases, it is difficult to follow which are the systems that have been simulated. A small table in the Supplementary Material containing the simulated systems with the key structural features (complete TS, helical lid, ...) may be helpful. I suggest also adding a reference to the Supplementary Methods Text and/or Table in the first paragraph of the results section to keep in mind which systems will be discussed in the following sections.

8. It is not specified between which atoms the distances are calculated. As has been done for the scissile phosphate - catalytic core distance and TS bending angle, it should be mentioned (in the captions or in the supplementary methods) between which atoms (or groups of atoms in case of center of mass) the selected distances (e.g. Figure 2 and Supp Fig. 1) have been measured (e.g. in the case of phosphates it is the phosphorus,

some oxygen or the center of mass of a group of atoms?).

9. In Supplementary Figure 1, R1014 and K1018 are shown in sticks as the key side chains that establish salt bridge interactions with TS phosphates. In Figure 2, R1016 and K1013 are highlighted to interact with crRNA and TS respectively. However, if the viewpoint is the same in both figures is it possible that K1013 should be K1018 in Figure 2? From this perspective it looks like the lysine is farther in the lid than R1016. If this is correct, some rephrasing will be required in the main text.

10. In Supplementary Figure 1 (as well as in other Figures) it is not clear what the two lines (time-evolution) of each plot represent (2 replicas or two distances?). This should be clarified in the caption.

11. Based on plots shown in Figure 2 and Supplementary Figure 3, the TS bending shows different behaviors depending on the structure of the lid (helical (tendency toward 40°) and unstructured (toward 110°)). Is this because the reference angle is different for the two systems? or because they follow different trends? Is this functionally relevant?

12. The authors mention in the Supplementary methods that the Amber ff19SB force-field is used. This force-field was used in combination with the TIP3P water model. In principle, ff19SB shows better behavior with the OPC water model (J. Chem. Theory Comput. 2020, 16, 1, 528–552). Since the results obtained are in line with experimental observations this may not be an issue here. Maybe as a future suggestion, it would be interesting to see the effect of using different water models/protein force-fields in these complex systems.

13. In Figure 4 it would be useful to highlight the scissile phosphate/cleavage site region.

14. In Supplementary Figure 9, I suggest adding information about which are the States A, B, and C in the plot. Also, a white solid line connecting the initial and final states is depicted. Is this the minimum free energy path? In this case, I recommend adding this information in the figure caption.

Reviewer #3 (Remarks to the Author):

I do not recommend this work for publication for two reasons.

1. The model of Cas12a after NTS cleavage (Fig. 1c) was obtained by considering two structures (PDB 6GTG and PDB 5NFV). Moreover, the positions of the two key magnesium ions were also modeled. Thus, it is likely that the initial model (Fig. 1c) biases the sampled configurations in the wild-type as well as the mutated cases. Moreover, even after running the long simulations, the scissile phosphate seems to be significantly far from the catalytic core of the RuvC domain. What is the distance between the scissile phosphate and the side chains of the catalytic core (D917, E1006, D1255) after MD? Thus, the conformational changes observed in the MD simulation do not necessarily lead to the catalytically active conformation. The observed conformational change might also suggest that the initial structural bias drives the system to some irrelevant local minima in the energy surface. The simulation starting with the PDB 5NFV with modeled lid (unresolved in the X-ray structure) also showed the appearance of salt bridge interactions (K1013-TS, R1016-crRNA, Supplementary Fig. 3). However, the salt bridges were unresolved in the X-ray structure containing full-length TS with the scissile phosphate.

2. The alchemical free energy estimates associated with the decoupling TS (38 nucleotides, Fig. 1c) in complex with protein (WT or mutant) is a very large chemical transformation and likely to be limited by sampling and convergence issues. 11 lamda

windows seem to be very very less. However, convergence has been claimed for ~22 ns alchemical simulations. It is unclear if every window is 22 ns. It is known the protein (in complex with crRNA) undergoes a large conformational change in response to DNA binding, which is unlikely to be sampled by the alchemical simulations. Thus, the estimated free energies are questionable.

The reported results might be biased by the initial structural models.

We thank the reviewers for their positive evaluation of our manuscript and for their constructive criticisms, which greatly helped us in improving our manuscript. We have responded to each point, made necessary changes in the manuscript, and added new analyses in the Supplementary Information (SI) as outlined below.

Reviewer #1 (Remarks to the Author):

1. Clarification required. Supp Fig 1 shows the distance between selected residues from the α -helical lid (R1014, K1013 and K1018) and either the crRNA or the TS. More information is required about how these distances were calculated – for instance, is it the minimum distance to any atom of the respective nucleic acid strand? Or the distance to the COM of the nucleic acid strand? This information is required to know how to interpret the distances. The generic nature of these distances is in contrast to those plotted in Fig 2, which are between K1013 to C2 of the TS and R1016 to U12 of the crRNA, and Supp Fig 3, where they are between K1013 and G1 of the TS and R1016 and U11 of the crRNA. Why calculate specific distances in some cases and generic distances in others? On what basis were the specific residue/base pairs selected?

We thank the reviewer for raising this important point, which requires clarification. We computed the distances between specific atoms when we intended to analyse the interactions between two residues (**Fig. 2a,b, Supplementary Fig. 1 and 4a,b**), and we considered COM distances to show the overall movement of a region of the complex with respect to another region (**Fig. 2d and Supplementary Fig. 4d**). To characterize the interactions between positively charged residues of the lid and the nucleic acids, we computed the time evolution of the distances between specific atoms, nitrogen atoms of the positive amino acid residues and negative oxygen atoms of the nucleic acid phosphate backbone for analysing interactions. We have now changed the figure legends (**Page 7, lines 163-166; Page S11, lines S309-312; Page S14, lines S353-357**) to clarify the calculation of the distances well.

In our study, we considered two experimentally resolved structures (PDB ID: 5NFV and 6GTG) of the *Francisella novicida* Cas12a. Owing to the differences in the starting structure, and the length of the TS, we have the same residues of the lid interacting with slightly different (adjacent bases) phosphate backbone of the crRNA:TS duplex. For this reason, the K1013 and R1016 are shown to interact with different crRNA and TS bases respectively in **Fig. 2 and Supplementary Fig. 4a**. We have also corrected **Supplementary Fig. 1** adding the specific bases of the crRNA:TS duplex interacting with the lid.

2. Fig 2, P6: It is noted more than once that “biochemical and single-molecule experiments have shown that the cleavage site on the TS can mainly be located between base positions -2 and -4.” (see also P10), yet only the distance between the catalytic core and the phosphate connecting the T-2 and G-3 nucleobases is calculated. Why not calculate the distance to the G-3 and A-4 phosphate too?

Biochemical and single-molecule experiments have indeed indicated that the scissile phosphate is between -2 and -4 positions of the TS (Cofsky et al. *Elife* **2020**, *9*, e55143). We considered P_{SCl} between -2 and -3 consistently throughout the length of our paper. This is because -2 and -3, and -3 and -4 being adjacent phosphates in the TS backbone show approximately similar behaviour in our MD simulations. We have now included an additional Supplementary Figure (**Supplementary Fig. 2d,e**) to report these data.

3. Fig 2, Supp Fig 1, 3, 4, 5, 7: The time-series data presented throughout this manuscript are shown as a darker line over a lighter line of the same colour. Presumably the darker line is a running average and the lighter line the raw data, but this needs to be stated somewhere (along with the averaging time period).

The reviewer is correct. The lighter translucent lines in the plots show the actual data points, while the solid lines show the running average over 5 ns windows of MD simulation. We have now included this information in

the figure legends (Page 8, lines 177-178; Page S11, lines S319-321; Page S12, lines S337-339; Page S14, lines S362-363; Page S15, lines S373-375; Page S16, lines S388-390; Page S17, lines S403-405; Page S19, lines S438-440; Page S23, lines S480-481; Page S24, lines S487-489; Page S25, lines S503-504).

4. In the interpretation of the time-series data presented in Fig 2 and Supp Fig 1 and 4, particular time points are noted as being when transitions occur (sometimes a dashed line is drawn, other times not, even when it is the same data presented, e.g. Fig 2c vs Supp Fig 4d). How were these time points identified? Simply by eye or through some sort of statistical analysis? The 6 μs time point at which stable salt bridges apparently form in Supp Fig 1b is clearly incorrect – the distances do not become short enough to represent salt bridge formation, nor stable, until close to 7 μs . The definition of “stable formation” of an interaction is also not provided.

We thank the reviewer for pointing out the need for clarification in our time evolution plots.

Supplementary Fig. 1b shows that salt bridge interactions form approximately 7 μs of MD onwards and persist under 4 Å through the remainder of the simulation, as a sign of stable interaction. Salt-bridge interactions are measured as oxygen-nitrogen distance and are considered formed under 4 Å (Kumar & Nussinov, *Biophys. J.* **2002**, 83, 1595–1612). We have corrected Supplementary Fig. 1 and its legend, reporting that the lid forms stable salt-bridge interactions with the DNA TS at approximately 7 μs (Page S11, lines S317-319). We also included the reference reported above.

Since we performed long timescale simulations (~10 μs for each replica), a vertical dashed line was used to indicate the approximate time of a substantial conformational change, e.g., when a marked drop in the time evolution plots is observed. We have now included this dashed line in Supplementary Fig. 6d the way it has been shown in Fig. 2c.

5. It is claimed in the caption of Supp Fig 1 that K1013 and R1014 remain linked to positions 8 and 12 of the crRNA backbone from 2 μs onwards, yet the distances clearly move beyond 5 Å, which would indicate they are not bound (although this depends on how the distance was calculated, see comments on Supp Fig 1 above) during the remainder of the simulation.

The reviewer correctly points out that K1013 and R1014 form intermittent interactions with the crRNA at positions 8 and 12, especially after ~4 μs . We have now corrected the figure legend (Supplementary Fig. 1; Page S11, lines S316-317).

6. Again, in the interpretation of Fig 3a and Supp Fig 4a, it is claimed that the decrease in the bending angle of REC2b “becomes more substantial after ~6 μs ”, but there are other times when large changes occur, e.g. just after 3 μs . Why focus on this one? Additionally, the time-series data of the two replicas split at this time point, with only one replica becoming more bent, but this is not mentioned in the text.

This is also correct. Noteworthy changes in the bending of the REC2b region are observed at both ~3 μs and ~6 μs (Supplementary Fig. 6a), with one replica changing more than the other. We have now incorporated this information in the main text (Page 8, lines 190-193).

7. Supplementary Methods, “Calculation of bending angles”: this section would benefit from reference to the various figures with schematics of these angles. However, even the schematics do not appear to represent what is described in the methods, or at least it is hard to relate the schematics to the methods, as the nucleotide bases are not numbered and the planes are not shown. A better schematic should be provided. Moreover, according to the method for constructing the vectors, the time-series should always start at 90°, yet many of

the time-series clearly do not start here (see e.g. Supp Fig 7a), so what was the reference structure? It needs to be clearly stated (and justified) in each case.

We would like to apologize for this plotting error in Supplementary Fig. 7a. The reviewer correctly points out that the angles for REC2b bending, duplex arching, and Nuc bending start at 90°. This is indeed reported in **Supplementary Fig. 6a-c** for the complete Cas12a system. We have now corrected the error in **Supplementary Fig. 9a** (previously Supplementary Fig. 7a). In all cases, the system was used upon energy minimization as a reference structure for the calculation of bending angles. We have now reported this information in the caption of **Supplementary Figs 5, 6 and 9 (Page S15, lines S368-370; Page S16, lines S386-387; Page S19, lines S423-425)**.

We made an informed decision toward not showing the plane for visual clarity and instead showed only the representative vector that is on the plane formed based on the selection of the region of interest. We computed the angle between this vector and its perpendicular over the course of our simulations. Our selection for the individual domains has been provided in **Supplementary Table 2**. We have rewritten the description in the Supplementary Methods for further clarification (**Pages S7-8, lines S216-230**).

8. Fig 2: Part (d) shows the distance between the scissile phosphate and the centre of mass of the RuvC catalytic core. Are the magnesium ions included in the COM calculation? This isn't clear from the text but the accompanying image (bottom left) would suggest they are.

Mg²⁺ ions were included while considering the COM of the RuvC catalytic core. We have now included this information in the legend of **Fig. 2d (Page 8, lines 172-174)**.

9. Supp Fig 2: What is the RMSD of the lid residues computed with respect to? Is this an all-atom or C α only RMSD?

We computed the all-atom root-mean-square deviation (RMSD) of the lid along MD simulations with respect to the structures obtained experimentally. Hydrogen atoms were excluded from the RMSD calculation. We have now included this information in the legend of now **Supplementary Fig. 3 (Page S13, lines S342-345)**.

10. Supp Fig 7: Part (b) is interpreted in the caption as showing that “the nucleobases containing the scissile phosphate, T-2 (RMSD < 1.5 Å) and G-3 (RMSD < 0.5 Å), report a remarkable stability.”, yet the RMSD for G-1 is very similar to that of T-2 and the RMSD of C-5 is lower than G-1 and T-2, so why is T-2 considered “remarkably stable”?

We apologize for the confusion. In this figure (now **Supplementary Fig. 9**), we evaluated the catalytically competent state of Cas12a where the TS locates within the active site. In this system, the scissile phosphate locates between the G₋₃ and T₋₂ bases and coordinates the Mg²⁺ ions (**Supplementary Fig. 9d**). We wanted to bring the focus to the base G₋₃, which displays remarkable stability. The flanking bases, including T₋₂, also show appreciably low RMSD, highlighting the stability of the TS at the active site. We have rephrased the legend of **Supplementary Fig. 9** to bring more clarity (**Page S19, lines S428-432**).

11. Supp Fig 7: What is the reference structure for the RMSD time-series shown in part (c)? It cannot be the initial structure as the time-series to not start from 0, yet the initial structure is what it should be for an interpretation in terms of stability during the simulation.

We thank the reviewer for bringing this to our notice. This was a plotting error as we skipped some of the initial frames by mistake. We have now corrected the data. We used the initial structure (i.e., the system upon energy minimization) as a reference to compute the RMSD time series in now **Supplementary Fig. 9c**.

12. What data support the statement “Accordingly, the K1206-R1218 α -helix/ β -sheet region of Nuc maintains persistent interaction with the 5'-tail of the DNA TS.”? There needs to be a reference to one or more figures.

We have now performed additional contact analysis, which we have included in **Supplementary Fig. 15e**. Our new analysis shows that the K1206-R1218 region of Nuc maintains its interaction with the 5'-tail of the DNA TS in both simulation replicas.

13. P18/SM P2: How were the missing residues in 5NFV constructed? Was a template used? Or structural constraints? Why did the reconstruction not produce a helical structure for the lid region?

The missing residues N1009 – F1017 in the 5NFV structure were reconstructed (**Fig. 1b**), along with other missing loops, through homology modelling using the SWISS-MODEL software by the Schwede group (Waterhouse et al. *Nucleic Acids Res.* **2018**, *46*, 296–303; Biasini et al. *Nucleic Acids Res.* **2014**, *42*, 252–258). The PDB: 5NFV was used as a template for the overall structure and conformation of the system. We have added this information under the “Structural Models” section of Supplementary Methods (**Page S2, lines S33-36**).

14. SM P2: Why wasn't the X-ray structure of CRISPR-Cas12b (5U30) able to show the active site and Mg²⁺ ions (for which an additional X-ray structure of CRISPR-Cas12i2 (6LTU) was required)?

The X-ray structure of CRISPR-Cas12b (5U30, solved at 2.92 Å resolution) (Yang et al. *Cell* **2016**, *167*, 1814–1828) was obtained in the absence of Mg²⁺ ions (by omitting Mg²⁺ during crystallization). This is a common experimental practice to inhibit nucleic acid cleavage, as a result of which the Mg²⁺ ions are absent from the active site. For this reason, the X-ray structure of CRISPR-Cas12i2 (6LTU, solved at 2.57 Å resolution), which includes Mg²⁺ ions within the active site, was used as a reference (**Supplementary Fig. 6**).

15. SM P2: How exactly was the structure of the catalytically-competent state of Cas12a built? What software was used? How were the different structural features combined?

This is indeed an important point that needs further clarification. The catalytically competent state of Cas12a for TS cleavage was built taking cues from related crystal structures of Cas12a orthologues, holding the DNA TS within the RuvC active site. The X-ray structure of the CRISPR-Cas12b (PDB: 5U30, at 2.9 Å resolution) accommodates the TS within the RuvC catalytic cleft. The TS bends down from the crRNA:TS hybrid toward the active site, forming a U-shaped strand with its 5'-tail interacting with the Nuc domain. Cas12a and Cas12b share a highly conserved RuvC catalytic domain, which is superposable in the 6GTG and 5U30 structures (**Supplementary Fig. 8b**). A comparison of the two structures shows that the nucleic acids are also superposable and differ for a longer U-shaped TS that locates with the RuvC active site in Cas12b. Building on this similarity, the 5U30 structure was used as a template for biased MD simulations, starting from the complete Cas12a system that includes Mg²⁺ ions in the RuvC site. Restraint forces were applied to reduce the RMSD between initial and final conformations of the DNA nucleobases -1 to -5, which include the scissile phosphate, as also shown in a prior modeled structure of Cas12a (Swarts et al. *Mol Cell* **2017**, *66*, 221-233.e4). A smooth transition was achieved in reducing the RMSD gradually over ~600 ns (details are now reported in the Supplementary Methods; **Page S3, lines S63-82**). The resultant structure contains the scissile phosphate, between the nucleobases T-2 and G-3, which coordinates the catalytic Mg²⁺ ions (**Supplementary Fig. 8c,d**). This complex was subjected to ~10 μ s of MD simulation for further refinement and to ensure long timescale stability (**Supplementary Fig. 9**). Along the simulations, the active site is stable (**Supplementary Fig. 9c**) and the DDE motif steadily coordinates Mg²⁺ (**Supplementary Fig. 9d**).

16. P18/SM P3: The Berendsen barostat does not produce a thermodynamically correct NpT ensemble. Was this barostat only used for the equilibration stages or also used during the production simulation?

We apologize for not clarifying the details explicitly in the methods. The Berendsen barostat was used only during the equilibration and thermalization phases of our conventional MD simulations (details in the *Methods* section and in the *Supplementary Methods*; **Page S4, lines S97-98**).

Long timescale simulations on Anton-2 were performed in the NPT ensemble. Pressure control was accomplished via the Martyna, Tobias, Klein (MTK) barostat, set to maintain 1 bar of pressure, with a tau (piston time constant) parameter of .0416667 ps and reference temperature of 310.15 K. The barostat period was set to the default value of 480 ps per timestep. Temperature control was accomplished via the Nosé-Hoover thermostat with a tau (time constant) parameter of .0416667 ps. We have now included these details in both the main text (**Pages 19-20, lines 490-496**) and in the *Supplementary Methods* (**Page S4, lines 117-123**), also including the appropriate references.

17. P20/SM P6: Were position restraints used in the alchemical free energy perturbation to prevent movement of the TS as its interactions were reduced?

In our alchemical free energy simulations, we decoupled only the specific bases of the TS (A_{14} , C_2 , G_1 and G_{-1}) that interact with the α -helical lid to compute their energetic contribution in binding the protein. The remaining regions of the TS were treated without any restraints, given that the crRNA-protein complex maintains steady contact with the TS. For instance, the Nuc domain establishes persistent contacts with the 5'-tail of the TS (**Supplementary Fig. 7b**). We have clarified this information in the *Supplementary Methods* (**Page S8, lines S233-237, 246**).

18. SM P4: In the calculation of RC1, was the $C\alpha$ RMSD calculated over all amino acids in Cas12a, or only some?

In the calculation of RC1, we used the RMSD of all the $C\alpha$ atoms. We have now explicitly reported this information in both the main text (**Page 11, line 257**) and the SI (**Page S4, line 175**).

Additional analysis required

19. P8/9: The claims of coupled motion of the REC2 and Nuc domains is in my opinion not fully justified. The presentation of the time-series data in Fig 3 is very difficult to interpret, with the diagonal offset of the time-series making it particularly hard to check the claimed correlation of the time-scales/patterns over time of the paired datasets (“similar trend”, “in concert with”). Why not present these data in regular time-series plots like in Fig 2 and many of the supplementary figures (including Supp Fig 5, referred to in this section)? Additionally, to better validate the claims of processes happening in concert, scatter-plots could be used and the correlation between the two properties could be calculated. Another possibility would be to use principle component analysis to identify the “large-scale coupled motions” supposedly exhibited by the major domains.

In **Fig. 3**, we reported the most salient data showing the cooperativity in the decrease of the REC2b bending and hybrid arching angles, as well as between the Nuc and TS bending angles. We thereby succinctly presented the data from one replica in **Fig. 3** and for two replicates in **Supplementary Fig. 6a-d**, in which regular time-series plots are reported. We have now clarified this point in the main text (**Page 8, lines 191-193 a**, and **Page 9, line 208-209**) and in the legend of **Fig. 3**. We have now also provided **Supplementary Fig. 6f-g**, reporting the positive correlation between the variables using scatter plots. The correlation observed between the pairs is substantial (i.e., Pearson’s correlation coefficient $r = 0.84$ between the hybrid arching and REC2 bending, and $r = 0.70$

between the Nuc and TS bending) considering that the data refer to the bending of individual domains. A comment on this positive cooperativity is reported in the main text (Page 8, lines 198-199, and Page 9, lines 213). Additionally, we performed principal component analysis (PCA) and reported the “essential dynamics” derived from the first principal component (PC1, Supplementary Fig. 6e). The arrows show the overall direction of motion, and the size of the arrows is proportional to the amplitude of motions of the individual residue.

Additional discussion required

20. P5: The interactions between Cas12 and the TS differ between the cryoEM and X-ray crystal structures. Can the authors please discuss these differences (which are presumably due to the different lengths of TS in each)?

The reviewer correctly points out that we performed MD simulations of both the 6GTG cryo-EM structure and the 5NFV X-ray structure, as well as of a more complete CRISPR-Cas12a system after NTS cleavage (Fig. 1a-c, Supplementary Table 1). In the 6GTG cryo-EM structure, the α -helical lid interacts with the crRNA through K1018. Over the course of our long timescale MD simulations, the α -helical lid interacts with the crRNA (binding U₁₂ and C₈) and anchors the TS by interacting with C₃ and T₄ (Supplementary Fig. 1).

In the 5NFV X-ray structure, which contains a longer TS, the lid was reconstructed as an unstructured loop using homology modelling. During MD simulations of this system, the loop lid interacts with G₁₁ of the crRNA and A₁ of the TS (Supplementary Fig. 4a,b), which are adjacent nucleobase positions compared to the 6GTG cryo-EM structure.

MD runs on the complete Cas12a system, based on the 6GTG cryo-EM structure and containing a longer TS, show that the α -helical lid interacts with U₁₂ of the crRNA and C₂ of the TS (Fig. 2a,b), similar to MD simulations of the 6GTG cryo-EM structure.

We have now clarified the interactions by precisely labelling the crRNA:TS bases in Fig. 2a,b and Supplementary Figs. 1 and 4a,b and discussed in the main text (Page 6, lines 153-155).

21. P6: “Interestingly, this unstructured lid forms salt-bridges with the crRNA and intermittent interactions with the TS (Supplementary Fig. 3a-b). This, in turn, affects the bending of the TS and the movement of the scissile phosphate toward the RuvC catalytic site (Supplementary Fig. 3c-d.” “affect” in what way? Supp Fig 3 does not show any clear trend in the TS bend angle, which remains > 90°, i.e., not really bent (cf angles around 40° that occur with the helical lid). On the other hand, the distance to the catalytic core seems to decrease faster for the unstructured lid compared to the helical lid and reaches similarly short distances. How does this happen without TS bending? These differences mean that the next sentence, “This indicates that the interactions of the lid with both the crRNA and TS are critical to achieving the bent conformation of the TS.” is hard to justify, as interactions occur with either lid structure but TS bending only occurs with the helical lid.

We thank the reviewer for bringing this important point to our notice. The reviewer correctly points out that the unstructured lid fails to establish persistent interactions with the TS (now Supplementary Fig. 4a-b). To further understand the observation of a decrease in the P_{SCI}–RuvC core distance, we computed the REC2b bending angle and the duplex arching (Supplementary Fig. 5a,b). We observe a strongly correlated decrease (Pearson’s correlation coefficient $r = 0.88$, Supplementary Fig. 5d) in these angles along our simulations, which likely suggests that the inward bending of REC2b and the duplex arching facilitate the nearing of P_{SCI} toward the RuvC active site (Supplementary Fig. 5c,e). This is indeed supported by single-molecule data, reporting that the high flexibility of REC2 could be crucial for DNA cleavage (REFs 12, 16, 20-23). This suggests that an unstructured lid is functionally less effective in aiding the traversal of the TS, than the α -helical lid structured in the PDB: 6GTG. In this respect, we recall that in the 5NFV structure, the lid was disordered and reconstructed here as a loop through homology modeling. We have now discussed this point in the text (Page 6, lines 151-160) and included our new analysis in Supplementary Fig. 5.

22. P12: The interactions between Cas12a and the TS observed in the unbiased and umbrella sampling simulations are briefly compared; I would like to see a larger discussion of the differences, especially given the apparently barrierless PMF between the state simulated in the unbiased simulations and State B observed in the umbrella sampling – can the authors comment on why this state did not occur in the unbiased simulations?

This is a very interesting aspect. Our free energy simulations reflected a barrierless transition from state A to state B, in which the α -helical lid actively participates in the traversal of the DNA TS toward the RuvC active site (**Fig. 4, Supplementary Fig. 13**). In our unbiased $\sim 10 \mu\text{s}$ -long classical MD simulations, where the α -helical lid interacts with the TS, we observed a $\sim 6 \text{ \AA}$ movement of the P_{SC1} toward the RuvC core (**Fig. 2**). However, the descent of the TS from state A to state B observed through free energy methods involved an $\sim 18 \text{ \AA}$ reduction in the P_{SC1} –RuvC core distance (**Supplementary Fig. 9**), which could not be captured through $\sim 10 \mu\text{s}$ -long MD simulations in replicates. We have now addressed this point in the Discussion (**Page 16, lines 401-406**).

23. P12/Fig 4: More discussion is required of the fact that the $\Delta\Delta G$ value for the N1009A mutation is large yet its effect on activity is relatively small.

Alchemical free energy simulations show that the $\Delta\Delta G$ value for N1009A is high in state A but drops to a moderate value in state B (**Fig. 4b**). In the DNA cleavage assay, we found that N1009A leads to a significant reduction in cleavage, but comparably less than F1012A, K1013A, and R1016A (**Fig. 4c**). We have included this observation in the discussion (**Page 16, lines 416-418**).

Labelling and presentation of figures

24. Fig 1: In part (a) of the figure, the domains are labelled but the REC and NUC lobes referred to in the text on P3 are not – which domains do they each contain? The PI domain labelled in the figure is not defined or mentioned anywhere in the manuscript. Lastly, Rec1 and Rec2 are not fully capitalised here but are in all subsequent figures.

We thank the reviewer for bringing this to our notice. The REC lobe is constituted of the REC1 and REC2 domains of Cas12a, while the NUC lobe contains the RuvC and Nuc domains (**Page 3, lines 57-59**). We have now also explicitly mentioned that the PAM-interacting (PI) domain is responsible for the recognition and binding of a short protospacer-adjacent motif (PAM) in the target DNA (**Page 3, lines 60-63**). We have capitalised the REC1 and REC2 labels in **Fig. 1**.

25. Supp Fig 1: The residues shown in the snapshots accompanying part (a) need to be labelled like they are for the snapshot accompanying (b). Also please check that it is K1013 and not K1018 shown in these snapshots (hard to tell based on the viewing angle).

We have now updated **Supplementary Fig. 1 (Page S11, lines S309-312)**.

26. Supp Fig 1: “R1014 and K1018 approach the DNA TS at positions 2 and 3 respectively” – isn’t it the opposite, i.e., R1014 approaches TS position 3 and K1018 approaches TS position 2? (at least that’s what it looks like from the snapshot accompanying part (b)).

We have corrected the caption of **Supplementary Fig. 1 (Page S11, lines S316-319)**.

27. Arrows are used in Fig 3 (and other figures) to indicate the bending of the REC2b and Nuc domains. Similar

arrows should be added for the hybrid arching and TS bending, which are even harder to identify in the snapshots.

We have now added the arrows in the snapshot of **Fig. 3**. Additionally, we have provided the essential dynamics captured by principal component 1 (PC1), showing the overall direction of motion of protein residues and nucleobases using arrows (**Supplementary Fig. 6e**).

28. Supp Fig 5: Part (d) does not add anything that cannot already be seen, more quantitatively, in parts (a-c). If you must keep it, it should be the same width as parts (a-c) so that comparisons between the panels can be easily made.

The reviewer pointed out correctly for which we have now removed the spectrogram representation of the contacts data, and the updated figure can be seen in **Supplementary Fig. 7**.

29. Supp Fig 8: Please label the various residues and nucleobases drawn in stick format.

We have added the labels for all the snapshots in this figure (**Supplementary Fig. 10**).

30. Fig 4: Please label the amino acid residues shown in the State C snapshots (and ideally also the nucleotides in all three snapshots).

We have labelled all the residues and bases shown in the snapshots of this figure (**Fig. 4**).

31. Fig 5: The REC domains could be more comprehensively labelled.

We have explicitly labelled all the domains in **Fig. 5a**.

Typographical errors and unclear phrasing

32. P5: “Indeed, the α -helical lid draws the TS toward itself by...” – this phrasing is confusing – is the TS being drawn toward itself (as written), or toward the α -helical lid (what I think is meant)?

We have adjusted the text for clarity (**Page 5, line 118**).

33. Fig 2 caption: “through the G-1, G1, C2, C3 and T4” – no “the” is required.

We have corrected the text (**Page 8, line 170**).

34. Supp Fig 5: “A contact is considered when the distance between two heavy atoms among the regions of interest is lesser than 3.5 Å.” – should just be “less”.

We have corrected the text (**Page S17, line S403**).

35. P14: What is meant by describing the RNA-bound Cas12a complex as “binary”?

CRISPR-Cas systems are usually called binary when the protein is complexed with the crRNA only (as mentioned in the main text, **Page 13, line 347-349**) and it is referred to as ternary when it is bound to both crRNA and DNA.

36. Fig 5: “establishing stable salt-bridge interactions: - should be “salt-bridge”

We have corrected the text (*Page 18, line 453*).

37. Supp Methods P4: “Restrain forces” \diamond should be “Restraint forces”

We have corrected the text (*Page S6, line S180*).

38. Supp Methods P5: “with a collective force constant k was 50” – I think “was” should be “of”?

We have corrected the text (*Page S6, line S190*).

Reviewer #2 (Remarks to the Author):

1. In PDB 5NFV and in the MD simulations based on this initial structure, the lid is unstructured and flexible. However, the structured alpha-helical lid is selected when building the catalytically competent Cas12a (Supplementary Figure 6) and in PDB 6GTG with the large TS (Figure 2), which are also the final and initial states of the umbrella sampling simulations, respectively. Therefore, the question is: which is the role of the disordered lid in the DNA traversal mechanism? or in other words, is the disordered lid functionally relevant? The authors should clarify this aspect in the manuscript.

The reviewer correctly points out that the lid was disordered in the X-ray structure 5NFV (**Supplementary Table 1**) and reconstructed as a loop by homology modelling (details in the Methods section). Long timescale simulations of this system show that the unstructured loop intermittently interacts with the TS and fails to bend the TS (**Supplementary Fig. 4c**), which was deemed crucial for the accommodation of the TS at the RuvC active site (Cofsky et al. *Elife* **2020**, 9, e55143). This suggests that an unstructured lid is functionally less effective in aiding the traversal of the TS, than the α -helical lid structured in the PDB: 6GTG. In light of these observations, we advanced our study to investigate the role of the α -helical lid. We have now clarified the section in the main text reporting the data from the simulations of the 5NFV X-ray structure (**Page 6, lines 149-160**).

2. Related to the previous question, is the alpha-helical lid changing its conformation/order when going from state A to C in the free-energy simulations? Besides the initial approach toward the crRNA:TS, in Figure 4 it looks like there are some structural changes during the process (which are more pronounced in the mutant, see state A in Supplementary Figure 8 which is partially unstructured). For example, is F1012 always pointing toward TS DNA or it depends on the order/disorder of the lid? It would be interesting to see if the helical character is particularly stabilized in a particular state along the traversal process.

We have now added **Supplementary Fig. 14** to comment on this observation in detail. Indeed, the lid loses its alpha helicity temporarily and restores the conformation again along the minimum free energy path (MEP) of the umbrella sampling simulations. This is observed in the wild-type system where the RMSD of the lid increases as the system enters state A and drops down to a stable value as the profile slides down toward state B, where it establishes an extensive network of interaction with the crRNA:TS duplex (**Supplementary Fig. 14a-c**). In the mutated system, where the interacting residues are mutated in alanine, the lid loses its alpha helicity transiently and restores it to a stable value through the rest of MEP (**Supplementary Fig. 14a,b,d**), most likely owing to alanine mutations that hold the maximum propensity toward alpha helicity (Pace et al. *Biophys. J.* **1998**, 75, 422–427). We have now added these discussions to the main text (**Page 11, lines 272-276; Page 12, lines 302-304**).

In the wild-type system, each of the positively charged residues (K1013, R1014, R1016, and K1018) along with N1009 and F1012 of the lid interact with several nucleobases of the TS over the MEP (**Supplementary Fig. 13**), capturing the descent of the TS toward the RuvC active site. In particular, F1012 progressively directs toward the nucleobase in the TS (G₋₁) and establishes stacking interactions (**Fig 4a, Supplementary Fig. 13b**) more prominently at state B. An analysis of the interactions is now reported in **Supplementary Fig. 13**. Details are also briefly reported in the main text (**Page 11, lines 268-272**).

3. In the first and second section of the results (i.e. helical lid anchor crRNA:TS and Rec2 and Nuc motions), a lot of emphasis is put on the some key distances and angles. However, these key features are not analyzed in the free energy simulations. I think that it would be interesting to analyze how these features/interactions change (if possible) along the minimum free-energy path (e.g. which interactions between the alpha-helical lid and TS are formed and broken or the process of TS bending). This will provide additional molecular information on the sequence of events depicted in Figures 4 and 5.

This is indeed an interesting point that adds to our study. We have now included new analyses reporting the interactions of the positively charged residues (K1013, R1014, R1016, and K1018), N1009 and F1012 with the crRNA:TS duplex (**Supplementary Fig. 13**). This analysis delineates the molecular details at the level of the interactions that the lid establishes with the crRNA:TS duplex. Additionally, we have also provided the changes in the major domains (i.e., REC2 and Nuc) at critical states along the minimum energy pathway, showing that the large domain motions remain unperturbed by the mutations in the lid (**Supplementary Fig. 15**). We briefly discussed these observations in the Discussion section (**Page 12, lines 300-302**).

4. One of the major findings is the cooperative motions between different structural elements that facilitate TS traversal (helical lid, Rec2, and Nuc). Based on these results and the scheme shown in Figure 5, first the alpha-helical lid approaches TS DNA and, then, the coupled motion of Nuc and Rec2 occurs. In a previous work (J. Chem. Inf. Model. 2020, 60, 12, 6427–6437), the authors explored the essential dynamics (using principal component analysis) of Cas12a in their DNA and RNA bound states. I wonder if performing a similar analysis for the MD simulations of 6GTG with the large TS may provide relevant information about the coupled motions of the three domains: helical lid, Rec2 and Nuc. Is the helical lid that triggers Rec2 and Nuc closure or all motions are in some way correlated? This should only be added in the Supplementary Material if it provides additional information on the potential cooperativity.

We have now also provided **Supplementary Fig. 6**, reporting the coupled REC2b bending and duplex arching along with Nuc bending and TS bending. We observed a positive correlation between the variables as shown using scatter plots (also suggested by Reviewer 1; **Supplementary Fig. 6e-g**). The correlation observed between the pairs is substantial (i.e., $r = 0.84$ between the hybrid arching and REC2 bending; $r = 0.70$ between the Nuc and TS bending) considering that the data refer to the bending of individual domains. A comment on this positive cooperativity is reported in the main text (**Page 8, lines 198-199, and Page 9, lines 212-213**). Additionally, we have also reported the “essential dynamics” of the complex captured by the first principal component (PC1) from principal component analysis (**Supplementary Fig. 6e**).

We do not have direct evidence to support the claim that the alpha helical lid triggers the motions of REC2 and Nuc, however, their motions are indeed correlated as shown in **Supplementary Fig. 6**.

5. How are the representative snapshots of States A, B, and C (shown in Figure 4 and also the starting point for TI simulations) obtained? Are they obtained based on visual inspection/manual selection or using some clustering algorithm? This should be specified in the Supplementary Methods.

We first obtained the trajectory along the minimum free energy path based on the values of the reaction coordinates (details in the Supplementary Methods, **Page S6, lines S174-180**). Next, we took snapshots of the representative states based on the reaction coordinates values along the minimum free energy path. We have now rephrased the captions of **Fig. 4** and **Supplementary Fig. 10** to clarify this (**Page 15, lines 364-366; Page S20, lines S447-448**). The starting points for TI simulations were also considered in the same manner, and this is now clarified in the Supplementary Methods (**Page S9, lines S261-262**).

6. In Figure 4c, the results of the in vitro DNA cleavage for the mutation E1006Q are shown but not discussed in the manuscript.

The E1006Q mutant is the catalytically inactive variant and was used as a negative control for the DNA cleavage assay. We have now mentioned this in the main text (**Page 12, lines 317-320**).

Additionally, I have some minor comments/suggestions about some aspects that need to be clarified:

7. In some cases, it is difficult to follow which are the systems that have been simulated. A small table in the Supplementary Material containing the simulated systems with the key structural features (complete TS, helical lid, ...) may be helpful. I suggest also adding a reference to the Supplementary Methods Text and/or Table in the first paragraph of the results section to keep in mind which systems will be discussed in the following sections.

We have now included a table summarizing the systems studied (*Supplementary Table 1*) and enlisted the salient features in them. The table is cited in the “Results” paragraph.

8. It is not specified between which atoms the distances are calculated. As has been done for the scissile phosphate - catalytic core distance and TS bending angle, it should be mentioned (in the captions or in the supplementary methods) between which atoms (or groups of atoms in case of center of mass) the selected distances (e.g. Figure 2 and Supp Fig. 1) have been measured (e.g. in the case of phosphates it is the phosphorus, some oxygen or the center of mass of a group of atoms?).

We thank the reviewer for raising this important point. We considered atomic distances when analysing the interaction between two residues and distances between the centre of mass (COM) of larger regions to describe their overall movements. We have now updated all figure legends in both the main text (*Fig. 2*) and the SI (*Supplementary Figs. 1, 4*), where information on distance analysis was missing. Details are also reported in the reply to reviewer 1, point 1.

9. In Supplementary Figure 1, R1014 and K1018 are shown in sticks as the key side chains that establish salt bridge interactions with TS phosphates. In Figure 2, R1016 and K1013 are highlighted to interact with crRNA and TS respectively. However, if the viewpoint is the same in both figures is it possible that K1013 should be K1018 in Figure 2? From this perspective it looks like the lysine is farther in the lid than R1016. If this is correct, some rephrasing will be required in the main text.

We apologize for this typographical error. *Supplementary Fig. 1* reports data from the simulation of the cryo-EM 6GTG structure, which lacks the region of the TS downstream of the crRNA:TS duplex. In this system, K1013 and R1016 interact with the TS, while R1016 and K1018 interact with crRNA. *Fig. 2* reports data from the complete Cas12a system, including a longer TS with the scissile phosphate. Here, K1013 interacts with the TS, and R1016 interacts with the crRNA. We have now corrected the residue labels in *Supplementary Fig. 1* and in its caption (*Page S11, lines S309-312*).

10. In Supplementary Figure 1 (as well as in other Figures) it is not clear what the two lines (time-evolution) of each plot represent (2 replicas or two distances?). This should be clarified in the caption.

The time-series data throughout the manuscript are shown as a solid line over a lighter line of the same colour (*Fig 2, Supplementary Figs 1, 2, 4-7, 9, 13-15*). The lighter translucent lines in the plots show the actual data points, while the solid lines show the running average over 5 ns windows of MD simulation. We have now included this information in the figure legends (*Page 8, lines 177-178; Page S11, lines S319-321; Page S12, lines S337-339; Page S14, lines S362-363; Page S15, lines S373-375; Page S16, lines S388-390; Page S17, lines S403-405; Page S19, lines S438-440; Page S23, lines S480-481; Page S24, lines S487-489; Page S25, lines S503-504*).

11. Based on plots shown in Figure 2 and Supplementary Figure 3, the TS bending shows different behaviors depending on the structure of the lid (helical (tendency toward 40°) and unstructured (toward 110°)). Is this because the reference angle is different for the two systems? or because they follow different trends? Is this functionally relevant?

We thank the reviewer for pointing out this error. We mistakenly chose a different selection mask for the calculation of the domain angles in the 5NFV X-ray structure, as compared to the complete Cas12a system. We have now replotted the data in **Supplementary Fig. 4c** based on the same selection mask as used for the complete Cas12a system (**Supplementary Table 2; Fig. 2c**). In all cases, the system upon energy minimization was used as a reference starting structure for the calculation of bending angles.

12. The authors mention in the Supplementary methods that the Amber ff19SB force-field is used. This force-field was used in combination with the TIP3P water model. In principle, ff19SB shows better behavior with the OPC water model (J. Chem. Theory Comput. 2020, 16, 1, 528–552). Since the results obtained are in line with experimental observations this may not be an issue here. Maybe as a future suggestion, it would be interesting to see the effect of using different water models/protein force-fields in these complex systems.

This is indeed a very interesting point. Tian and co-workers suggest that the combination of ff19SB with the OPC four-point explicit water model (Izadi et al. J. Phys. Chem. Lett. 2014, 5, 3863–3871) should have better predictive power for modelling proteins in solution and for rational protein design. We chose the TIP3P water model primarily for the relatively higher speed of MD computations for a large biomolecular system like CRISPR-Cas12a (~307,000 atoms). Due to the extra point for the description of water (despite three in TIP3P), the performance most likely drops by as roughly adding 1 extra atom per water molecule. However, we agree with the reviewer that it would be interesting to study the effect of different force-field models on the dynamics of protein/nucleic acid complexes, a goal that is definitively part of our future research.

13. In Figure 4 it would be useful to highlight the scissile phosphate/cleavage site region.

We have now labelled the scissile phosphate of the TS.

14. In Supplementary Figure 9, I suggest adding information about which are the States A, B, and C in the plot. Also, a white solid line connecting the initial and final states is depicted. Is this the minimum free energy path? In this case, I recommend adding this information in the figure caption.

The white solid line represents the minimum free energy path connecting the initial and final states. We have updated the figure (now **Supplementary Fig. 11**) and its caption with the information on the states (**Page S21, lines S458-460**).

Reviewer #3 (Remarks to the Author):

1. The model of Cas12a after NTS cleavage (Fig. 1c) was obtained by considering two structures (PDB 6GTG and PDB 5NFV). Moreover, the positions of the two key magnesium ions were also modeled. Thus, it is likely that the initial model (Fig. 1c) biases the sampled configurations in the wild-type as well as the mutated cases.

We built the complete Cas12a system (Fig. 1c) starting from the cryo-EM structure of Cas12a (PDB: 6GTG, Fig. 1a) that contains the α -helical form of the lid (L1008 – K1021). Since 6GTG lacks the 5'-tail of the TS (bases from -1 to -7) downstream of the crRNA:TS duplex, containing the scissile phosphate (Cofsky *et al.*, *Elife*, 2020, 9, e55143), we built it from the X-ray structure of Cas12a (PDB: 5NFV, Fig. 1b) that contains a longer TS. In detail, we performed an all-C α based RMSD fitting of the 6GTG structure on 5NFV (RMSD = ~ 2 Å, Supplementary Fig. 2a). Next, we performed a separate backbone based RMSD fitting (RMSD = ~ 2.5 Å) of the nucleic acids between the two structures and patched the TS:NTS hybrid downstream of the crRNA:TS duplex from the 5NFV structure onto 6GTG. We maintained the nucleic acid sequence of the 6GTG cryo-EM structure. This complex was further refined using a multi-template mode of the MODELLER program (Fiser & Sali *Methods in Enzymology* 2003, 374, 461–491). We have now added these details in the Supplementary Methods (page S2, lines S37-S46).

The RuvC domain of Cas12a is highly conserved across nuclease enzymes (Yang W. Q. *Rev. Biophys.* 2011, 44, 1–93) and holds a two-metal aided architecture, characterized by the highly conserved DEDD (or DDE) motif (Swarts *et al.* *Mol. Cell* 2017, 66, 221–233), in which Mg²⁺ ions are coordinated by carboxylate groups. The X-ray structure of Cas12i2 (PDB: 6LTU), is phylogenetically related to Cas12a, captured the catalytic Mg²⁺ ions coordinating with the DED motif in its RuvC domain (Supplementary Fig. 2b). The two catalytic Mg²⁺ ions were placed in the complete Cas12a system by superimposing the DDE motif in the RuvC domain between the two structures (6GTG and 6LTU). Finally, the complete Cas12a system was subjected to AMBER force field-based vacuum minimization to obtain an energetically stable complex, further used for long timescale MD simulations. Importantly, Supplementary Fig. 2c shows the stability of the DDE motif coordinating with the Mg²⁺ ions along ~ 10 μ s-long MD simulation, preserving the typical two-metal ion architecture. These findings strongly support the reliability of our simulated systems. We have now included this discussion in the Supplementary Methods (page S2, lines S46-S57) for clarification.

Finally, we note that we also performed independent MD simulations of the cryo-EM (PDB: 6GTG, Fig. 1a) and X-ray structure (5NFV, Fig. 1b) of the Cas12a system. Findings from the simulations of both the structures (Supplementary Fig. 1, Supplementary Fig. 4) converged well with the finding from the complete Cas12a system (Fig. 2, 3), thereby underscoring the reliability of the inferences drawn in this study.

2. Moreover, even after running the long simulations, the scissile phosphate seems to be significantly far from the catalytic core of the RuvC domain. What is the distance between the scissile phosphate and the side chains of the catalytic core (D917, E1006, D1255) after MD? Thus, the conformational changes observed in the MD simulation do not necessarily lead to the catalytically active conformation.

We thank the reviewer for giving us the opportunity to clarify that we have simulated four different systems: (1) the cryo-EM structure PDB: 6GTG (Fig. 1a), (2) the X-ray structure PDB: 5NFV (Fig. 1b), (3) a more complete system of Cas12a after NTS cleavage, based on the PDB: 6GTG and including a longer DNA TS (Fig. 1c), and (4) a catalytically competent Cas12a (Supplementary Fig. 8). To clarify this, we have included a new Supplementary Table 1, summarizing the simulated systems and their characteristics.

In the complete Cas12a system, the distance between the scissile phosphate (P_{ScI}) of the TS and the centre of mass of the RuvC core (D1255, D917, and E1006, coordinating with the two catalytic Mg²⁺ ions) is ~ 30 Å, consistent with the 5NFV structure. Our ~ 10 μ s-long MD simulation of the complete Cas12a system showed a

reduction of ~ 6 Å between the P_{SCI} and the RuvC core (**Fig. 2d**). The reviewer thereby correctly points out that the complete conformational change is not observed through long timescale MD simulations. The descent of the TS toward the RuvC core involved an ~ 18 Å reduction in the P_{SCI}–RuvC core distance, and was described through free energy methods (**Fig. 4** and **Supplementary Fig. 9**).

In the catalytically competent Cas12a, the TS locates within the RuvC core and the P_{SCI} coordinates the Mg²⁺ ions (**Supplementary Fig. 8**). Long timescale MD simulations of this system reveal that the RuvC catalytic core is remarkably stable (**Supplementary Fig. 9c**), and the P_{SCI} coordinates the two Mg²⁺ ions, alongside the stable coordination of the DDE motif (**Supplementary Fig. 9d**). These findings strongly support the reliability of our simulated systems.

Long timescale simulations of the complete Cas12a system after NTS cleavage revealed the crucial players that can orchestrate the traversal of the TS toward the RuvC active site. Specifically, an α -helical lid was found to anchor the crRNA:TS duplex, while the TS bends and the scissile phosphate moves toward the RuvC core (**Fig. 2**). Concurrently, the REC2 and Nuc domains bend toward the enzymatic core, facilitating the traversal of the TS toward the catalytic pocket (**Fig. 3**). These observations prompted us to perform umbrella sampling simulations to investigate the traversal of the TS toward the RuvC catalytic core. In these simulations, the distance between the P_{SCI} and the RuvC core reduces from ~ 30 Å to ~ 6 Å (similar to what was observed in the X-ray crystal structure of Cas12i2; PDB: 6LTU).

3. The observed conformational change might also suggest that the initial structural bias drives the system to some irrelevant local minima in the energy surface.

This is an important observation that we also considered in our study. We agree with the reviewer on the concern regarding possible biases arising from the initial structure. That is why, we simulated two experimentally resolved structures, viz., 6GTG (cryo-EM at 3.27 Å resolution; **Supplementary Fig. 1**) and 5NFV (X-ray at 2.5 Å resolution; **Supplementary Fig. 4**) for ~ 10 μ s, alongside the complete Cas12a system (**Fig. 2**) in independent replicates. The major findings identified the same critical players taking part in the traversal of the TS and converged toward similar conclusions. Hence, our observations are reliable in terms of explaining the dynamics of Cas12a involved with the TS traversal.

4. The simulation starting with the PDB 5NFV with modeled lid (unresolved in the X-ray structure) also showed the appearance of salt bridge interactions (K1013-TS, R1016-crRNA, **Supplementary Fig. 3**). However, the salt bridges were unresolved in the X-ray structure containing full-length TS with the scissile phosphate.

The Reviewer correctly points out that, during long timescale simulations of the 5NFV structure of Cas12a from *Francisella novicida* post NTS cleavage, the unstructured lid forms transient interactions with the crRNA:TS duplex (**Supplementary Fig. 4**). 6GTG structure of the same Cas12a protein post NTS cleavage, in which the lid was indeed captured displaying an α -helical form, showed that the α -helical lid interacts with the crRNA through K1018. Molecular simulations of the 6GTG structure also reveal that the α -helical lid interacts and anchors the crRNA:TS duplex (**Supplementary Fig. 1**). We have used both the structures (6GTG and 5NFV) in our study, both of which showed salt bridge formation between the lid and the crRNA:TS duplex in the long timescale MD simulation. Furthermore, our analyses of the lid along the minimum free energy path of the Umbrella sampling simulations revealed the lid to be highly dynamic during guiding the TS toward the RuvC active site (**Supplementary Fig. 14**). Taken together, our findings suggest that the lid of Cas12a is inherently a structurally dynamic element of the protein, and its positively charged residues are poised to interact with the nucleic acids.

5. The alchemical free energy estimates associated with the decoupling TS (38 nucleotides, **Fig. 1c**) in complex

with protein (WT or mutant) is a very large chemical transformation and likely to be limited by sampling and convergence issues. 11 lambda windows seem to be very very less. However, convergence has been claimed for ~22 ns alchemical simulations. It is unclear if every window is 22 ns. It is known the protein (in complex with crRNA) undergoes a large conformational change in response to DNA binding, which is unlikely to be sampled by the alchemical simulations. Thus, the estimated free energies are questionable.

In our alchemical free energy simulations, we have decoupled only the specific bases of the TS (A₁₄, C₂, G₁ and G₋₁) that interact with the α -helical lid, to compute their energetic contribution in binding the protein. The remaining regions of the TS were treated without restraints since the TS maintains steady contact with the crRNA-bound complex.

We agree with the reviewer that, increasing the number of lambda windows could further fine-tune our data for which we recomputed the binding free energies with 21 lambda windows now (see Supplementary Methods, **Page S8, lines 238-240, Fig. 4b, Supplementary Fig. 16**). Our simulations, which we ran until ~30 ns for each window, sufficiently converged at ~22 ns (**Supplementary Fig. 12b**) and the overall inference of the relative binding free energy is in line with the experimental results (DNA cleavage assay, **Fig. 4c**). Based on these observations, the calculated binding free energies are reliable and explain the experimental findings.

Finally, the reviewer correctly points out that protein undergoes large conformational changes. For this reason, we performed alchemical free energy simulations considering two conformational checkpoints (state A and state B, **Supplementary Fig. 8**) obtained from the minimum free energy path of umbrella sampling simulations.

6. The reported results might be biased by the initial structural models.

This is an important observation that we have considered and addressed in the reply to Reviewer 3, point 3.

Reviewer #1 (Remarks to the Author):

For the most part, the authors have done a fantastic job of responding to my suggestions. I have just a few remaining points, which I have numbered according to their rebuttal, as they are all points I didn't think were fully addressed.

4. The authors have not explained how the substantial conformational changes marked by dashed vertical lines were identified. For instance, in Supp Fig 6a, the dark line undergoes what looks like a major transition at about 1 μ s, but only the transitions just after 3 μ s and at 6 μ s are identified with dashed lines. What were the criteria for designating a transition as major?

5. I appreciate the authors acknowledging the instability of salt bridge formation. However, their revision isn't entirely correct – in 1a, K1018 forms very consistent salt-bridge interactions, it's only K1013 whose interactions are intermittent. Additionally, in 1b, while the running average is always below 4 Å, the instantaneous distances for both K1013 and R1016 are occasionally above 4 Å, so strictly speaking these salt bridge interactions are also at least somewhat intermittent, although so few disruptions probably still counts as stable salt bridge formation.

7. The authors state here that the reference system was the energy-minimised structure but in the revised Supplementary Methods section "Calculation of bending angles" they state "We generated a plane based on the first frame of reference (i.e., the cryo-EM/X-ray structure)". Which is correct?

Reviewer #2 (Remarks to the Author):

The authors significantly improved the manuscript and clarified the different aspects following the suggestions pointed out by the reviewers. In particular, it is interesting to see the evolution of the lid secondary structure and key interactions along the minimum energy path of the DNA traverse, which seems to confirm that the unstructured lid may be functionally less effective. Overall, the molecular insights provided using computational simulations on the complex DNA traverse mechanism in Cas12a pave the way toward the application of this methodology to further understand and engineer genome-editing in Cas12a.

I only suggest some minor aspects:

1. It would be useful to specify which non-bonded model has been used to describe Mg²⁺ ions. Li and Merz proposed various 12-6 and 12-6-4 non-bonded models for divalent ions.

2. The temperature at which the umbrella sampling simulations have been performed should be specified. Considering that different temperatures have been used for conventional MD (300 K) and Anton-2 simulations (310.15 K), the authors should mention which temperature is used for the free energy calculations.

Reviewer #5 (Remarks to the Author):

First, this paper represents a novel and detailed DNA cleavage mechanism by FnCas12a, as follows: after NTS cleavage, the alpha helical lid motif forms a salt bridge with the crRNA-TS heteroduplex to stabilize the crRNA-TS heteroduplex. Subsequently, in concert with the alpha helical lid salt bridge formation, the REC2b domain in concert with the salt bridge formation of the alpha helical lid, which bends the TS at the end of the crRNA-TS heteroduplex to the RuvC active site. Then, the Nuc domain also undergoes a conformational change that closes the crRNA-TS heteroduplex to hold the TS stable at the RuvC active site. TS is stably held in the RuvC active site by inducing a conformational change that also closes the Nuc domain. This mechanism is in stark contrast to the previous hypothesis of the more simple TS cleavage mechanism based on the crystal structure and cryoEM structure: Nuc induces TS to the RuvC active site, and Nuc contributes to the stabilization of TS induced to RuvC.

Second, the model of TS cleavage proposed in the paper seems reasonable enough, since the interaction formation between the alpha helical lid and crRNA-TS and the conformational change such as the closure of the REC2 domain are interactions also seen in other Cas12, and the model proposed in this study seems to work to some extent for Cas 12 as a whole. On the other hand, although the structural change has been known for some time, it has not been considered as a necessary step for TS cleavage, and therefore, this study may have shown a new significance of the structural change between lid and REC2. Only concern is that the Urea PAGE results for the DNA cleavage assay in vitro are quite dirty, so there is room for improvement there.

Six Review3 comments seem to boil down to a question that the MD results due to an initial model-dependent bias that the authors gave, and do not universally explain the Cas12a TS cleavage mechanism. In the viewpoint of structural biology, I think the results of this MD are quite reasonable, so I don't think we need to consider the bias issue. Also, in response to the above reviewer comment, the authors claim that the results of this MD are not local minima due to the initial model bias, since they turned the MD with multiple pdb of FnCas12a registered in pdb as the initial model, and they all obtained similar results. I feel this is actually the best argument the authors can make, so I think the CONCERN in review3 is well addressed.

Reviewer #1

For the most part, the authors have done a fantastic job of responding to my suggestions. I have just a few remaining points, which I have numbered according to their rebuttal, as they are all points I didn't think were fully addressed.

We want to thank the reviewer for their detailed inspection of the paper and for helping us in improving the manuscript significantly. Here are our responses to the remaining doubts:

4. The authors have not explained how the substantial conformational changes marked by dashed vertical lines were identified. For instance, in Supp Fig 6a, the dark line undergoes what looks like a major transition at about 1 μ s, but only the transitions just after 3 μ s and at 6 μ s are identified with dashed lines. What were the criteria for designating a transition as major?

The reviewer correctly pointed out that the vertical lines in Supp Fig. 6a are not based on a consistent metric. These vertical lines were placed based on visual inspection of the trajectory and the plots. We thereby decided to remove them from the figures to avoid confusion.

5. I appreciate the authors acknowledging the instability of salt bridge formation. However, their revision isn't entirely correct – in 1a, K1018 forms very consistent salt-bridge interactions, it's only K1013 whose interactions are intermittent. Additionally, in 1b, while the running average is always below 4 Å, the instantaneous distances for both K1013 and R1016 are occasionally above 4 Å, so strictly speaking these salt bridge interactions are also at least somewhat intermittent, although so few disruptions probably still counts as stable salt bridge formation.

The reviewer pointed out correctly that indeed the interactions for K1013 and R1016 intermittently go beyond the accepted limit (4 Å) of salt bridge interactions. The participating residues maintain persistent electrostatic interactions during the collected dynamics. We have now made necessary modifications in the manuscript, calling such interactions as “electrostatic interactions”.

7. The authors state here that the reference system was the energy-minimised structure but in the revised Supplementary Methods section “Calculation of bending angles” they state “We generated a plane based on the first frame of reference (i.e., the cryo-EM/X-ray structure)”. Which is correct?

We want to thank the reviewer for pointing out the error and for the confusion caused. We have always used the first frame of MD production runs as the frame of reference and have made necessary corrections in the Supplementary Methods.

Reviewer #2

The authors significantly improved the manuscript and clarified the different aspects following the suggestions pointed out by the reviewers. In particular, it is interesting to see the evolution of the lid secondary structure and key interactions along the minimum energy path of the DNA traverse, which seems to confirm that the unstructured lid may be functionally less effective. Overall, the molecular insights provided using computational simulations on the complex DNA traverse mechanism in Cas12a pave the way toward the application of this methodology to further understand and engineer genome editing in Cas12a.

We thank the reviewer for their comments and acknowledge the importance of our findings, reporting that the alpha-helical lid is a critical functional unit in Cas12a.

I only suggest some minor aspects:

1. It would be useful to specify which non-bonded model has been used to describe Mg²⁺ ions. Li and Merz proposed various 12-6 and 12-6-4 non-bonded models for divalent ions.

We used the 12-6 model of non-bonded interactions for Mg²⁺ ions as proposed by Li and Merz. We have now clarified this aspect in the main text and in the Supplementary Information.

2. The temperature at which the umbrella sampling simulations have been performed should be specified. Considering that different temperatures have been used for conventional MD (300 K) and Anton-2 simulations (310.15 K), the authors should mention which temperature is used for the free energy calculations.

We ran the Umbrella Sampling simulations at 300 K. We have now clarified this aspect in the main text and in the Supplementary Information.

Reviewer #5

First, this paper represents a novel and detailed DNA cleavage mechanism by FnCas12a, as follows: after NTS cleavage, the alpha-helical lid motif forms a salt bridge with the crRNA-TS heteroduplex to stabilize the crRNA-TS heteroduplex. Subsequently, in concert with the alpha-helical lid salt bridge formation, the REC2b domain in concert with the salt bridge formation of the alpha-helical lid, which bends the TS at the end of the crRNA-TS heteroduplex to the RuvC active site. Then, the Nuc domain also undergoes a conformational change that closes the crRNA-TS heteroduplex to hold the TS stable at the RuvC active site. TS is stably held in the RuvC active site by inducing a conformational change that also closes the Nuc domain. This mechanism is in stark contrast to the previous hypothesis of the more simple TS cleavage mechanism based on the crystal structure and cryoEM structure: Nuc induces TS to the RuvC active site, and Nuc contributes to the stabilization of TS induced to RuvC.

We thank the reviewer for their very positive evaluation of our manuscript and for highlighting the novelty of our mechanistic insights.

Second, the model of TS cleavage proposed in the paper seems reasonable enough, since the interaction formation between the alpha helical lid and crRNA-TS and the conformational change such as the closure of the REC2 domain are interactions also seen in other Cas12, and the model proposed in this study seems to work to some extent for Cas 12 as a whole. On the other hand, although the structural change has been known for some time, it has not been considered as a necessary step for TS cleavage, and therefore, this study may have shown a new significance of the structural change between lid and REC2. Only concern is that the Urea PAGE results for the DNA cleavage assay in vitro are quite dirty, so there is room for improvement there.

We thank the reviewer for the criticism concerning the Urea PAGE gels. We agree that the fluorescently labelled target strand (TS) DNA oligonucleotide, as used for the nuclease activity assays and visualized in the Urea PAGE, contains detectable amounts of abortive synthesis products and free fluorescent dye. The oligo was sourced in the HPLC-purified form, as now specified in the methods section, suggesting that there may have been a vendor QC issue. We reasoned that substrate oligo could nevertheless be used for our analysis, since the same dsDNA substrate was used consistently for all the different FnCas12a mutant proteins. The kinetics of TS DNA cleavage was quantified densitometrically, measuring the rate of loss of the substrate (i.e. full-length) band. The background signals therefore do not affect the quantification of the band, and the effect of contamination by the abortive synthesis products can be considered negligible.

Six Review3 comments seem to boil down to a question that the MD results due to an initial model-dependent bias that the authors gave, and do not universally explain the Cas12a TS cleavage mechanism. In the viewpoint of structural biology, I think the results of this MD are quite reasonable, so I don't think we need to consider the bias issue. Also, in response to the above reviewer comment, the authors claim that the results of this MD are not local minima due to the initial model bias, since they turned the MD with multiple pdb of FnCas12a registered in pdb as the initial model, and they all obtained similar results. I feel this is actually the best argument the authors can make, so I think the CONCERN in review3 is well addressed.

We deeply appreciate the reviewer's comment. In response to the concerns of Reviewer 3, we described the minute details of our approach in building the catalytically active Cas12a system that is poised to cleave the TS.

Reviewer #1 (Remarks to the Author):

I am satisfied that the authors have now responded to all of my suggestions, and commend them on an excellent manuscript!

Reviewer #2 (Remarks to the Author):

The authors have done a great job clarifying and addressing all the concerns raised by the reviewers. From my point of view, the manuscript is ready for publication.

Reviewer #5 (Remarks to the Author):

Since the authors made reasonable rebuttals to my concerns, this reviewer is mostly satisfied and recommends publication in Nature Communications.